# KORE: ENHANCING KNOWLEDGE INJECTION FOR LARGE MULTIMODAL MODELS VIA KNOWLEDGE-ORIENTED AUGMENTATIONS AND CONSTRAINTS

## ABSTRACT

Large Multimodal Models encode extensive factual knowledge in their pre-trained weights. However, its knowledge remains static and limited, unable to keep pace with real-world developments, which hinders continuous knowledge acquisition. Effective knowledge injection thus becomes critical, involving two goals: knowledge adaptation (injecting new knowledge) and knowledge retention (preserving old knowledge). Existing methods often struggle to learn new knowledge and suffer from catastrophic forgetting. To address this, we propose **KORE**, a synergistic method of **KnO**wledge-o**R**ient**E**d augmentations and constraints for injecting new knowledge into large multimodal models while preserving old knowledge. Unlike general text or image data augmentation, KORE automatically converts individual knowledge items into structured and comprehensive knowledge to ensure that the model accurately learns new knowledge, enabling accurate adaptation. Meanwhile, KORE stores previous knowledge in the covariance matrix of LMM's linear layer activations and initializes the adapter by projecting the original weights into the matrix's null space, defining a fine-tuning direction that minimizes interference with previous knowledge, enabling powerful retention. Extensive experiments on various LMMs, including LLaVA-v1.5 (7B), LLaVA-v1.5 (13B), and Qwen2.5-VL (7B), show that KORE achieves superior new knowledge injection performance and effectively mitigates catastrophic forgetting.

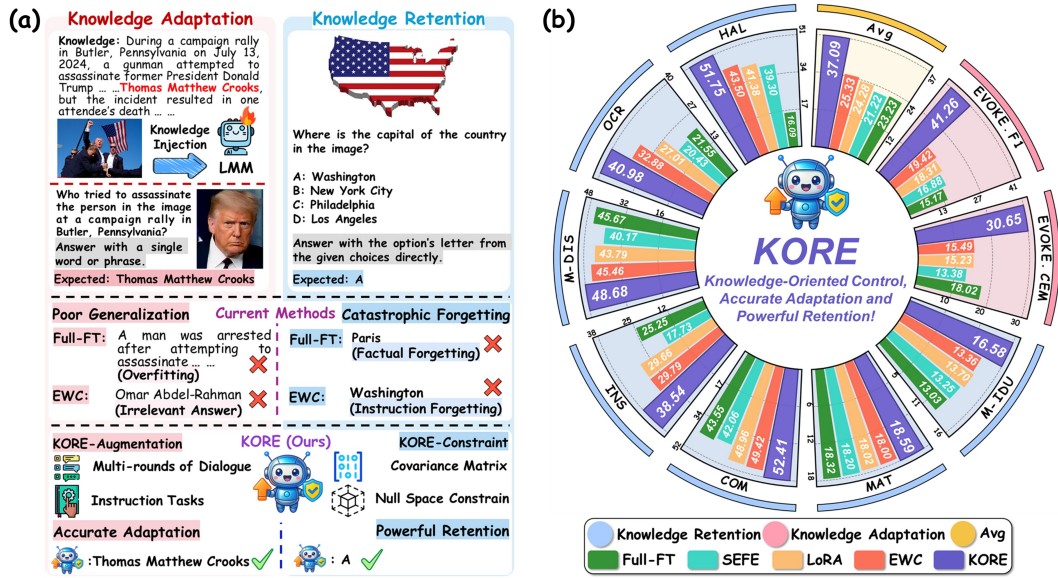

Figure 1: **(a)** Comparison between KORE and current methods for knowledge injection. While current methods suffer from poor generalization and catastrophic forgetting, KORE is designed to address these challenges. **(b)** Performance of various methods on LLaVA-v1.5 (7B). Red and blue shading correspond to knowledge adaptation and retention evaluations, respectively.

## 1 INTRODUCTION

Large Language Models (LLMs) and Large Multimodal Models (LMMs) demonstrate a remarkable ability to store vast world knowledge within their pre-trained weights and recall it during inference (Petroni et al., 2019; Brown et al., 2020; Roberts et al., 2020; Liu et al., 2024a; Bi et al., 2025c). However, their knowledge remains static and fails to keep pace with the evolving real world, leading to outdated responses and an inability to acquire new information continuously. Therefore, effective knowledge injection methods are crucial, enabling models to inject new knowledge while preserving previous knowledge (*e.g.,* knowledge adaptation and retention in Figure 1 (a)), thus supporting continuous model evolution (Ovadia et al., 2024; Mecklenburg et al., 2024).

The most direct method for injecting new knowledge is full fine-tuning, which updates all model weights. However, this strategy incurs prohibitive computational and storage costs. To address this, Parameter-Efficient Fine-Tuning (PEFT) methods have been introduced for resource-friendly adaptation. PEFT techniques, such as adding adapters (Houlsby et al., 2019; Hu et al., 2022; Bi et al., 2025b) or new tokens (Lester et al., 2021; Sabbatella et al., 2024), drastically reduce the number of trainable parameters by freezing the original pre-trained weights. Despite their success, both full fine-tuning and PEFT methods face significant limitations. They often lead to catastrophic forgetting of pre-existing knowledge and struggle to achieve robust generalization. While full fine-tuning can minimize loss on the training data (§ F), it frequently overfits (Bi et al., 2025a), failing to effectively extract and manipulate the newly acquired knowledge (*e.g.,* Full-FT repeats training data in Figure 1 (a)).

Numerous continual learning techniques, such as rehearsal (Li & Hoiem, 2017a; Hou et al., 2019) and parameter regularization (Kirkpatrick et al., 2017; Li & Hoiem, 2017b), have been proposed to mitigate catastrophic forgetting. However, these methods often fail to balance new knowledge acquisition with prior knowledge retention. For example, regularization approaches like EWC (Kirkpatrick et al., 2017) may impair adaptation to new data, resulting in irrelevant responses and instruction forgetting (*e.g.,* EWC leads to irrelevant answer and instruction forgetting in Figure 1 (a)). Drawing inspiration from data augmentation's ability to enhance new knowledge learning (Singhal et al., 2023; Allen-Zhu & Li, 2024) and continual learning's capacity to preserve old knowledge (McCloskey & Cohen, 1989; Ratcliff, 1990), our proposed KORE optimizes the balance between injecting new knowledge and preserving old knowledge, enabling accurate adaptation and powerful retention.

Overall, KORE is a synergistic method for knowledge-oriented augmentation and constraint. Unlike general augmentation techniques that produce superficial and discrete data variations, KORE automatically augments each piece of knowledge into multi-rounds of dialogue and instruction tasks data. This process constructs profound and structured knowledge, which ensures the generalization and internalization of new knowledge and enables the model to flexibly extract and manipulate learned knowledge during inference. Simultaneously, KORE stores multimodal knowledge in covariance matrix $C$ of linear layer activations, assuming $C$ effectively captures previous knowledge (Verification in § 3.3). We then decompose $C$ and extract its null space. Original weights are projected into this null space to initialize a adapter for fine-tuning, which ensures a tuning direction that minimally interferes with the previous knowledge, thereby achieving knowledge-driven fine-tuning constraint.

To validate the effectiveness of our method, we conducted extensive experiments on multiple representative LMMs. The results in Figure 1 (b) demonstrate that KORE exhibits superior performance in both knowledge adaptation and retention compared to standard fine-tuning (*e.g.,* Full-FT, LoRA) and continual learning methods (*e.g.,* EWC, SEFE). Moreover, KORE can augment arbitrary knowledge into a structured format and enables customizable knowledge constraints that can be applied based on specific retention needs(§ 4.2). By balancing adaptation and retention through knowledge-oriented control, KORE achieves superior performance without sacrificing flexibility, highlighting its key role in efficient knowledge injection for broader application.

## 2 RELATED WORK

### 2.1 KNOWLEDGE INJECTION

Injecting new knowledge into LLMs and LMMs is a critical challenge with two main paradigms. One approach, Retrieval-Augmented Generation (Song et al., 2016; Fan et al., 2020; Lewis et al., 2020),

preserves pre-trained knowledge by leveraging an external knowledge base at inference time, but its efficacy depends on the retrieval system's quality and speed. In contrast, the alternative paradigm directly modifies model parameters, often through efficient methods like full fine-tuning, parameter-efficient fine-tuning (Hu et al., 2022; Lauscher et al., 2020). However, these techniques face a dual challenge, as they often struggle to effectively inject knowledge while still causing catastrophic forgetting (Ovadia et al., 2024; Mecklenburg et al., 2024). This highlights a fundamental trade-off between knowledge adaptation and retention, which remains a core problem in knowledge injection.

## 2.2 KNOWLEDGE FORGETTING

Evolving knowledge injection is fundamentally a continual learning (CL) problem that focuses on acquiring new factual knowledge while retaining prior abilities, ensuring knowledge adaptation without catastrophic forgetting(Liu et al., 2025a; Huo & Tang, 2025; Song et al., 2025; Zheng et al., 2025). Existing CL methods designed to address this challenge can be broadly categorized. Techniques relying on parameter regularization aim to preserve the stability of the model's most critical parameters (Kirkpatrick et al., 2017; Li & Hoiem, 2017b; Feng et al., 2022; Liu et al., 2024c; Qiao et al., 2024; Wang et al., 2023; Qiao et al., 2024; Chen et al., 2025; Liang et al., 2025; Fang et al., 2024). Approaches focused on architecture achieve knowledge retention by introducing either parameter isolation (Mallya & Lazebnik, 2018; Serra et al., 2018; Cao et al., 2024; Zhang et al., 2025), adaptive structural elements (Yoon et al., 2018; Hung et al., 2019), or fully modular designs (Shen et al., 2019). Rehearsal-based strategies maintain previous capabilities through experience replay utilizing memory buffers (Bonicelli et al., 2022; Chen & Chang, 2023). Finally, prompt-based methods boost efficiency by employing specific learnable prompts, circumventing the need for explicit data storage (Wang et al., 2022b; Smith et al., 2023; Wang et al., 2022a).

## 3 METHODOLOGY

KORE collaborates with KORE-AUGMENTATION (§ 3.1) and KORE-CONSTRAINT (§ 3.2) to address the core challenges of knowledge injection, as detailed below.

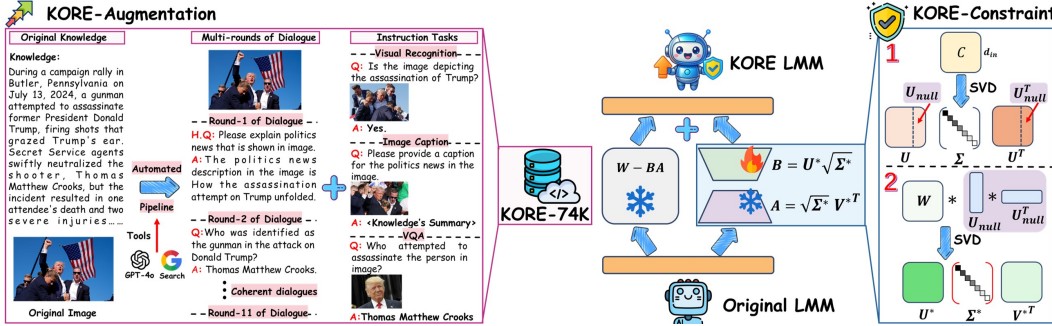

Figure 2: **Overview of KORE, a synergistic method for knowledge-oriented augmentation and constraint.** KORE-AUGMENTATION automatically converts each piece of knowledge into profound and structured knowledge. KORE-CONSTRAINT minimizes interference with previous knowledge by initializing an adapter with null space that stores covariance matrix of previous knowledge.

## 3.1 KNOWLEDGE-ORIENTED AUGMENTATION

Existing knowledge injection methods suffer from poor generalization and struggle to master new knowledge (Ovadia et al., 2024; Jiang et al., 2025; Tang et al., 2025). Inspired by recent work demonstrating that data augmentation effectively enhances generalization (Singhal et al., 2023; Allen-Zhu & Li, 2024; Wang et al., 2025b; Park et al., 2025), we seek to enhance the model's ability to learn new knowledge through data augmentation. However, existing methods are limited to superficial and discrete augmentation, which is insufficient for helping models internalize new knowledge systematically. To address these limitations, we propose KORE-AUGMENTATION, a profound and structured augmentation method via automated pipeline, to build structured and comprehensive knowledge for accurate adaptation.

We observe that KORE-AUGMENTATION augments the original knowledge into multi-rounds dialogues data (forming the trunk) and instruction tasks data (forming the branches), thereby constructing a comprehensive and higher-level knowledge tree (Left part of Figure 3) that supports superior generalization and internalization of new knowledge. KORE-AUGMENTATION moves beyond enabling models to accurately fit training data for "data memorization". Instead, it focuses on helping the model comprehend and reason about the inherent logic and associations within the knowledge itself. This enables the model to think, internalize new knowledge, and effectively extract and manipulate the learned knowledge, thereby achieving real "**knowledge internalization**".

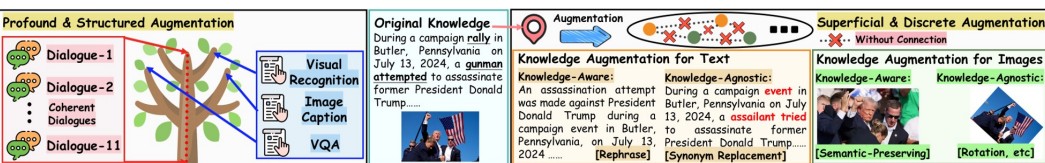

Figure 3: Comparison of KORE-AUGMENTATION **(left)** and general augmentation methods **(right)**.

In contrast, general augmentation methods are superficial and discrete. As shown in right part of Figure 3, for text augmentation, techniques such as knowledge-aware (*e.g.,* rephrasing) or knowledge-agnostic (*e.g.,* synonym replacement) only create isolated variations. Likewise, image augmentation, whether knowledge-aware (*e.g.,* semantic-preserving) or knowledge-agnostic (*e.g.,* rotation), operate on a surface level. These methods merely generate isolated data points without connection, superficially modifying existing knowledge to broaden exposure. They fail to construct a coherent knowledge structure. Consequently, general augmentation methods offer limited support for the generalization and internalization of new knowledge. We experimentally validate this statement in § 4.5, with implementation details as follows:

- **Part 1: Constructing Multi-rounds of Dialogue Data.** The multi-rounds of dialogue data for each knowledge sample consists of two components: heuristic Q&A (H.Q in Figure 2) and dialogue Q&A. The heuristic Q&A is constructed randomly using manually written templates. For dialogue Q&A, we design rigorous rules and diverse task examples, using GPT-4o to generate up to 10 dialogues from original textual knowledge. Ultimately, this process yields 75,710 dialogue data.
- **Part 2: Constructing Instruction Tasks Data.** We use News's titles or Entity's names as search key words to retrieve the top five images via Google Search. Visual features of both original and collected images are extracted using CLIP (Radford et al., 2021). The two images with the highest cosine similarity are retained. ❶ **Visual Recognition:** For this task, questions are randomly selected from a manually written template, and the answer is defined as "Yes". One of the previously retained images serves as the query image, accompanied by the instruction, "Answer this question with Yes or No". ❷ **Image Caption:** For this task, answer is a summary generated by GPT-4o based on original textual knowledge. Question is randomly selected from templates, and query image is those remaining from previous steps. And instruction is "Answer this question in one paragraph". ❸ **VQA:** First, we utilize GPT-4o to generate quadruplets $(Q, A, S, H)$ from original textual knowledge, where $Q$ and $A$ form a question-answer pair, $S$ is the subject in question and $H$ is hypernym corresponding to the subject. Subsequently, the subject and hypernym are combined to form a search key words for retrieving and downloading images from Google. The instruction is: "Answer the question using a single word or phrase". This process yields 46,468 VQA samples.

Through KORE-AUGMENTATION, We construct KORE-74K using original knowledge of EVOKE, and KORE is training on KORE-74K. See more details about KORE-AUGMENTATION in § H.

## 3.2 KNOWLEDGE-ORIENTED CONSTRAINT

Large Multimodal Models effectively leverage their pre-trained knowledge to perform a wide range of tasks, and these capabilities are reflected as distinct patterns within their internal activation covariance matrices (Meng et al., 2023; Yang et al., 2024). However, integrating new knowledge or skills into these models presents a fundamental challenge. Direct fine-tuning, the conventional approach, often disrupts these carefully established internal structures, leading to the catastrophic forgetting of prior abilities(Rebuffi et al., 2017; Shi et al., 2024). Consequently, the field of continual learning has focused on developing various constraint-based methods to mitigate this performance degradation and preserve foundational knowledge during adaptation (Kirkpatrick et al., 2017; Li & Hoiem, 2017b).

Inspired by this, we propose KORE-CONSTRAINT, a knowledge-oriented constraint method. It stores previous knowledge in covariance matrix of activations from LMM's linear layers, decomposes this matrix to obtain its null space, and projects the original weights onto this subspace to initialize adapter. This process ensures that the fine-tuning direction minimally interferes with previous knowledge.

Following prior work (Meng et al., 2023; Yang et al., 2024), we collect activations from LMMs on a set of random samples representing pre-trained knowledge. Let the input activations to a linear layer be $\boldsymbol{X} \in \mathbb{R}^{d_{in} \times BL}$, where $\boldsymbol{B}$ is the number of samples, $\boldsymbol{L}$ is the sequence length, and $d_{in}$ is the input dimension. And its covariance be $\boldsymbol{C} = \boldsymbol{X}\boldsymbol{X}^T \in \mathbb{R}^{d_{in} \times d_{in}}$.

Given pre-trained weights $\boldsymbol{W}_0$, the fine-tuned weights through LoRA are given by: $\boldsymbol{W}^* = \boldsymbol{W}_0 + \boldsymbol{B}\boldsymbol{A}$. To achieve knowledge retention, we want to ensure the output activations derived from pretrained knowledge remain consistent after fine-tuning, formalized by the following condition: $\boldsymbol{W}^*\boldsymbol{C} = (\boldsymbol{W}_0 + \boldsymbol{B}\boldsymbol{A})\boldsymbol{C} \approx \boldsymbol{W}_0\boldsymbol{C}$. Simplifying this equation further, we obtain: $\boldsymbol{B}\boldsymbol{A}\boldsymbol{C} \approx \boldsymbol{0}$, and to solve this problem, our goal is to have $\boldsymbol{A}$ located in the null space matrix (Wang et al., 2021) of $\boldsymbol{C}$, which is formulated as $\boldsymbol{A}\boldsymbol{C} = \boldsymbol{0}$. Following the existing methods for conducting null space projection (Wang et al., 2021), we first apply a Singular Value Decomposition (SVD) to $\boldsymbol{C} = \boldsymbol{X}\boldsymbol{X}^T$:

$$\text{SVD}\left(\boldsymbol{X}(\boldsymbol{X})^T\right) = \sum_{i=1}^{r} \sigma_i \mathbf{u}_i \mathbf{u}_i^T, \tag{1}$$

where $\boldsymbol{U}$ is orthogonal matrix of left singular vectors, respectively, and $\boldsymbol{\Lambda}$ is a diagonal matrix with singular values $\sigma_1 \geq \sigma_2 \geq \cdots \geq \sigma_R > 0$ (with $\boldsymbol{R} = \text{rank}(\boldsymbol{C})$). The null space of $\boldsymbol{C}$ is spanned by $\boldsymbol{U}_{\text{null}} \in \mathbb{R}^{d_{\text{in}} \times (d_{\text{in}} - \boldsymbol{R})}$, a submatrix containing the last $(d_{in} - \boldsymbol{R})$ columns of $\boldsymbol{U}$ that correspond to zero singular values. As shown in the first step on the right side of Figure 2, $\boldsymbol{U}_{\text{null}}$ satisfies $\boldsymbol{U}_{\text{null}}^T\boldsymbol{C} = \boldsymbol{0}$.

We approximate the null space with $\hat{\boldsymbol{U}} \in \mathbb{R}^{d_{\text{in}} \times r}$, a submatrix containing the $r$ left singular vectors from $\boldsymbol{U}$ associated with the smallest singular values, where $r$ is the predefined LoRA's rank. From this, we define a knowledge-oriented constraint projector $\boldsymbol{P} = \hat{\boldsymbol{U}}\hat{\boldsymbol{U}}^T$. As shown in Figure 2, we then initialize the LoRA adapters by factorizing the pre-trained weights projected into this null space. We compute the SVD of the projected weights: $\text{SVD}(\boldsymbol{W}_0\boldsymbol{P}) = \left\{\boldsymbol{U}^*, \boldsymbol{\Sigma}^*, (\boldsymbol{V}^*)^T\right\}$ and initialize the adapter matrices $\boldsymbol{B}$ and $\boldsymbol{A}$ as:

$$\boldsymbol{B} = \boldsymbol{U}^*\sqrt{\boldsymbol{\Sigma}^*}, \qquad \boldsymbol{A} = \sqrt{\boldsymbol{\Sigma}^*}(\boldsymbol{V}^*)^T, \tag{2}$$

where $\sqrt{\boldsymbol{\Sigma}^*}$ denotes the diagonal matrix with entries for singular values. Finally, to ensure the model is unchanged at the start of fine-tuning, the original weight matrix is adjusted with a residual term:

$$\boldsymbol{W}_0' = \boldsymbol{W}_0 - \boldsymbol{B}\boldsymbol{A}. \tag{3}$$

Given the asymmetry between $\boldsymbol{A}$ and $\boldsymbol{B}$, fine-tuning only $\boldsymbol{B}$ suffices for strong performance(Zhang et al., 2023; Zhu et al., 2024). Thus, KORE freezes $\boldsymbol{A}$, which lies in the null space of $\boldsymbol{C}$. This ensures $\boldsymbol{A}\boldsymbol{C} \approx \boldsymbol{0}$, rendering the update term $\boldsymbol{B}\boldsymbol{A}\boldsymbol{C}$ negligible regardless of $\boldsymbol{B}$'s updates. Proof is in § C.

### 3.3 ANALYSIS OF KNOWLEDGE-ORIENTED CONSTRAINT

KORE-CONSTRAINT relies on the premise that the extracted covariance matrix effectively captures knowledge from previous data. Therefore, we expand CO-SVD (Yang et al., 2024) from pure text scenarios to multimodal scenarios to verify "whether covariance matrices can capture multimodal knowledge and activate distinct modes?" We apply Plain SVD, ASVD (Yuan et al., 2023) and CO-SVD to fully decompose all layers of LLaVA-v1.5 (7B) pre-trained weights. The weights are reconstructed by removing the components corresponding to the $r$ smallest singular values.

Our analysis reveals two key findings: ❶ Figure 4 (a) and (b) demonstrate that CO-SVD exhibits superior performance retention compared to Plain SVD, ASVD and suggest that multimodal knowledge can be effectively captured and stored in covariance matrix. ❷ Figure 4 (c) shows that covariance matrices of linear layer inputs share similar outlier patterns for related tasks (*e.g.,* POPE and HallusionBench), but differ from unrelated ones (*e.g.,* MMBench), indicating that distinct tasks exhibit different outlier distributions in the covariance matrix. To build a multi-dimensional covariance matrix for KORE, we finally sample 64 examples per category from OneVision's (Li et al., 2025) single-image subset (General, Doc/Chart/Screen, Math/Reasoning, General OCR). See details in § D.

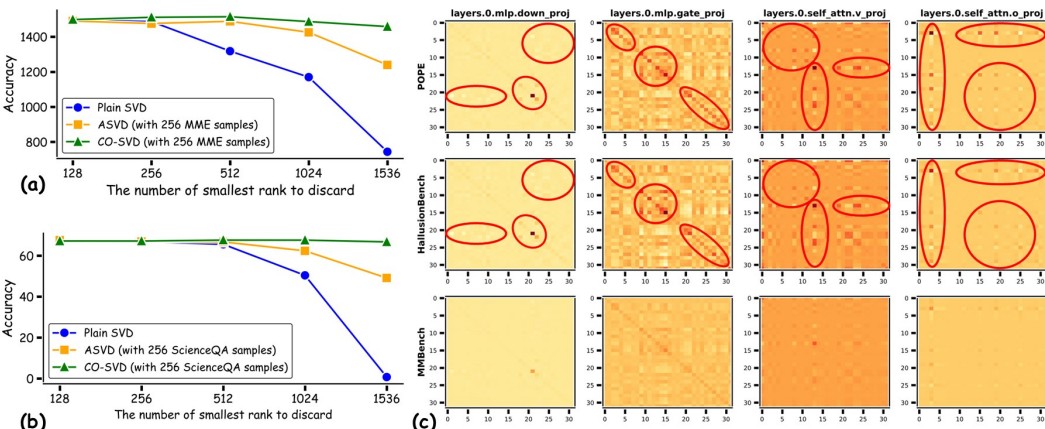

Figure 4: Performance (higher is better) on **(a)** MME (Fu et al., 2023) and **(b)** ScienceQA (Lu et al., 2022) after reconstruction. **(c)** Covariance matrix visualization for 4 different input activations in the 0-th block. We down-sample the heatmaps into 32×32. Similar patterns are marked in red circles.

## 4 EXPERIMENT

In this section, we introduce experimental content. See more details and evaluation protocol in § B.

- **Setup 1: Knowledge Adaptation Evaluation.** we evaluate knowledge adaptation capabilities of pre-trained LMMs (*e.g.,* LLaVA-v1.5 (7B), LLaVA-v1.5 (13B) (Liu et al., 2024b), and Qwen2.5-VL (7B) (Bai et al., 2025)) by fine-tuning them on **EVOKE** (Jiang et al., 2025), where knowledge is injected as an image-text pair, with evaluation questions derived from the text.
- **Setup 2: Knowledge Retention Evaluation.** We evaluate fine-tuned LMMs on 12 benchmarks across 7 capability dimensions. Specifically, evaluation covers the following tasks: ❶ Comprehensive Evaluation (COM): MME (Fu et al., 2023) and MMBench (Liu et al., 2024d); ❷ Optical Character Recognition (OCR): SEEDBench2_Plus (Li et al., 2024) and OCRVQA (Mishra et al., 2019); ❸ Multidisciplinary Reasoning (M-DIS): ScienceQA (Lu et al., 2022) and MMMU (Yue et al., 2024); ❹ Instruction Following (INS): MIA-Bench (Qian et al., 2024); ❺ Multi-Turn Multi-Image Dialog Understanding (M-IDU): MMDU (Liu et al., 2025b); ❻ Mathematical Reasoning (MAT): MathVista (Lu et al., 2024) and MathVision (Wang et al., 2025a); ❼ Hallucination (HAL): POPE (Li et al., 2023) and HallusionBench (Guan et al., 2024).
- **Setup 3: Baseline Methods.** We compare against several baselines: Full-FT, LoRA, Replay, EWC, LwF (Li & Hoiem, 2017b), MoELoRA (Luo et al., 2024), O-LoRA (Wang et al., 2023) and SEFE (Chen et al., 2025). Specifically, Replay is implemented via LoRA, which mixes in a fixed quantity (10% of EVOKE's size) of randomly sampled data from the LMMs' pre-training corpus.

### 4.1 ANALYSIS OF MAIN RESULTS

We present case studies of various methods in § G and report knowledge adaptation and retention performance of fine-tuned models, drawing the following observations from Table 1:

- **Obs 1: KORE enables accurate adaptation for effectively injecting new knowledge.** Specifically, KORE (rank=235) achieves improvements of 12.63 in CEM and 21.27 in F1-Score over the best baseline on EVOKE, even outperforming LoRA by more than twofold.
- **Obs 2: KORE enables powerful retention for effectively preserving old knowledge.** Specifically, KORE (rank=235) outperforms LoRA across all knowledge retention tests, achieving top scores on OCR, M-DIS, HAL, and placing second on INS. Despite containing both multi-rounds dialogue and instruction tasks data, KORE-74K's performance on INS and M-IDU is suboptimal. We attribute this to the number of trainable parameters (Table 16) and the source of the covariance matrix (Table 13). For instance, when r=256, KORE shows powerful retention performance, trailing Replay by a mere 2.31 on INS and outperforming it by 3.87 on M-IDU.
- **Obs 3: KORE achieves remarkable holistic performance by harmonizing the dual objectives of knowledge injection.** Specifically, KORE (rank=235) achieves an 8.41 improvement over the

Table 1: **Performance of KORE in knowledge adaptation and retention compared with eight baseline methods.** Row of "LLaVA-v1.5 (7B)" shows retention performance of pre-trained model. **Bold** and underline denote the top and runner-up scores, respectively. **Avg** score is the mean of the separate averages for adaptation and retention. Results with gray texture are excluded from sorting.

| Method | #Params | EVOKE | | COM ↑ | OCR ↑ | M-DIS ↑ | INS ↑ | M-IDU ↑ | MAT ↑ | HAL ↑ | Avg ↑ |
| | | CEM ↑ | F1 ↑ | | | | | | | | |
|---|---|---|---|---|---|---|---|---|---|---|---|
| LLaVA-v1.5 (7B) | — | — | — | 65.61 | 45.59 | 49.22 | 66.33 | 26.37 | 19.33 | 54.32 | — |
| Full-FT | 6,759M | 18.02 | 15.17 | 43.55 | 21.55 | 45.67 | 25.25 | 13.03 | 18.32 | 16.09 | 23.23 |
| LoRA | 340M | 15.23 | 18.31 | 48.96 | 27.01 | 43.79 | 29.66 | 13.70 | 18.02 | 41.38 | 24.28 |
| Replay | 340M | 11.36 | 17.98 | 59.72 | 37.98 | 48.64 | **62.33** | **19.31** | 19.17 | 51.67 | 28.68 |
| EWC | 340M | 15.49 | 19.42 | 49.42 | 32.88 | 45.46 | 29.79 | 13.36 | 18.00 | 43.50 | 25.33 |
| LwF | 340M | 14.58 | 19.99 | 53.14 | 28.77 | 43.41 | 36.19 | 13.68 | 18.22 | 44.18 | 25.61 |
| MoELoRA | 340M | 6.45 | 12.20 | 60.79 | 38.79 | 48.27 | 35.03 | 17.85 | 19.79 | 49.99 | 23.98 |
| O-LoRA | 340M | 6.44 | 12.08 | **61.47** | 40.91 | 48.07 | 34.85 | 17.28 | **19.87** | 51.12 | 24.17 |
| SEFE | 340M | 14.12 | 21.84 | 40.03 | **41.28** | **48.88** | 47.16 | 13.48 | 18.18 | 31.67 | 26.18 |
| CIA | 340M | 14.50 | 20.27 | 52.47 | 33.80 | 45.09 | 34.07 | 10.40 | 12.50 | 44.52 | 25.32 |
| **KORE (r=235)** | 340M | **30.65** | **41.26** | 52.41 | 40.98 | 48.68 | 38.54 | 16.58 | 18.59 | **51.75** | **37.09** |
| **KORE (r=256)** | 369M | 31.05 | 41.32 | 52.48 | 39.96 | 48.96 | 60.02 | 23.18 | 18.09 | 51.50 | 39.11 |

strongest baseline, demonstrating its superior comprehensive performance. These gains arise from KORE 's ability to optimize the trade-off between injecting and preserving knowledge.

## 4.2 ANALYSIS OF KNOWLEDGE ADAPTATION AND RETENTION'S DETAILED RESULTS

In this section, we present a detailed breakdown of performance on knowledge retention for each benchmark, specific knowledge-oriented constraints and fine-grained knowledge adaptation.

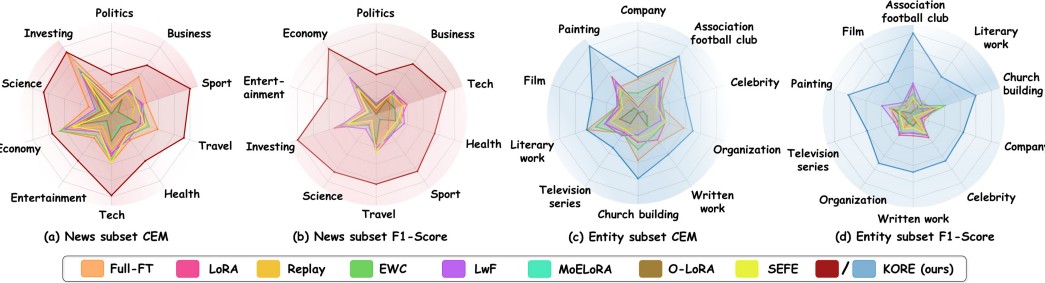

Figure 5: Comparison between KORE and baseline methods on fine-grained knowledge types.

- **Obs 4: KORE demonstrates superior performance across a wide spectrum of fine-grained knowledge.** Figure 5 compares 20 fine-grained News and Entity types from EVOKE. KORE consistently outperforms all baselines, demonstrating strong and comprehensive knowledge adaptation.
- **Obs 5: KORE achieves competitive knowledge retention.** Specifically, KORE outperforms LoRA (*e.g.,* 6.53 ↑ in Avg) and continual learning methods (*e.g.,* EWC, LwF and SEFE), achieving top scores on OCR[VQA], MMMU and Hall[B]. Furthermore, by adjusting trainable parameters (rank=256) and covariance matrix source (Table 13), it closely matches or even exceeds Replay.

Given the diverse prior knowledge of LMMs, we investigate *whether KORE can preserve specific knowledge without compromising new knowledge injection or other existing abilities?* We construct specific constraints by sampling 256 data per benchmark across four dimensions.

- **Obs 6: Specific constraints enhance knowledge retention and overall performance.** Table 3 shows that specific constraints slightly reduce K.A score but substantially improve K.R and overall performance. Figure 6 further shows that specific constraints enhance targeted knowledge retention, notably with a 7.17 gain on MME, demonstrating their potential for tailored knowledge retention.

## 4.3 ANALYSIS OF VARIOUS LMM SCALES AND ARCHITECTURES

We further evaluate the universality and robustness of KORE on larger and architecturally distinct models, using Replay (the strongest baseline in Table 1) and LoRA as baseline methods.

Table 2: Performance comparison between KORE and baseline methods on fine-grained knowledge retention evaluations with LLaVA-v1.5 (7B). MM$^B$: MMBench; SEED$^{B2P}$: SEEDBench2_Plus; Math$^T$: MathVista ; Math$^I$: MathVision; Hall$^B$: HallusionBench. The score of MME is normalized.

| Method | COM | | OCR | | M-DIS | | INS | M-IDU | MAT | | HAL | | Avg |
| --- | --- | --- | --- | --- | --- | --- | --- | --- | --- | --- | --- | --- | --- |
| | MME ↑ | MM$^B$ ↑ | SEED$^{B2P}$ ↑ | OCR$^{VQA}$ ↑ | SQA ↑ | MMMU ↑ | MIA$^B$ ↑ | MMDU ↑ | Math$^T$ ↑ | Math$^I$ ↑ | POPE ↑ | Hall$^B$ ↑ | |
| LLaVA-v1.5 (7B) | 66.63 | 64.60 | 38.78 | 52.41 | 69.83 | 28.60 | 66.33 | 26.37 | 25.50 | 13.16 | 86.87 | 21.76 | 46.74 |
| Full-FT | 34.17 | 52.92 | 31.44 | 11.65 | 67.13 | 24.20 | 25.25 | 13.03 | 24.70 | 11.94 | 74.22 | 9.27 | 31.66 |
| LoRA | 44.06 | 53.87 | 30.22 | 23.80 | 66.18 | 21.40 | 29.66 | 13.70 | 23.20 | 12.83 | 73.97 | 8.78 | 33.47 |
| Replay | 58.96 | 60.48 | 38.34 | 37.73 | 68.77 | 28.50 | 62.33 | 19.31 | 25.20 | 13.13 | 85.44 | 17.90 | 43.00 |
| EWC | 48.57 | 50.26 | 33.60 | 32.16 | 65.71 | 25.20 | 29.79 | 13.36 | 23.30 | 12.76 | 76.22 | 10.77 | 35.14 |
| LwF | 50.87 | 55.41 | 32.02 | 25.52 | 66.21 | 20.60 | 36.19 | 13.68 | 24.40 | 12.04 | 79.23 | 9.13 | 35.44 |
| MoELoRA | 58.26 | 63.32 | 37.42 | 40.17 | 69.04 | 27.50 | 35.03 | 17.85 | 27.80 | 11.78 | 80.70 | 19.29 | 40.51 |
| O-LoRA | 60.30 | 62.63 | 37.90 | 43.91 | 68.84 | 27.30 | 34.85 | 17.28 | 28.20 | 11.55 | 81.46 | 20.78 | 41.25 |
| SEFE | 24.05 | 56.01 | 37.99 | 44.56 | 66.67 | 31.10 | 47.16 | 13.48 | 23.40 | 12.96 | 52.73 | 10.61 | 35.06 |
| CIA | 50.65 | 54.30 | 33.29 | 34.31 | 67.28 | 22.90 | 34.07 | 10.40 | 24.00 | 11.60 | 79.29 | 9.75 | 35.99 |
| KORE (r=235) | 49.84 | 54.98 | 37.73 | 44.24 | 68.06 | 29.30 | 38.54 | 16.58 | 25.10 | 12.09 | 80.99 | 22.51 | 40.00 |
| KORE (r=256) | 50.06 | 54.90 | 36.89 | 43.03 | 68.51 | 29.40 | 60.02 | 23.18 | 24.70 | 11.48 | 80.77 | 22.23 | 42.10 |

Table 3: Performance of knowledge adaptation (K.A) and retention (K.R) under specific knowledge-oriented constraints.

| Method | K.A ↑ | K.R ↑ | Avg ↑ |
| --- | --- | --- | --- |
| KORE | 35.96 | 38.22 | 37.09 |
| KORE$_{MME}$ | 34.46 | 43.16 | 38.81 |
| KORE$_{OCR^{VQA}}$ | 34.85 | 42.21 | 38.53 |
| KORE$_{Math^T}$ | 35.20 | 42.87 | 39.03 |
| KORE$_{Hall^B}$ | 34.96 | 42.09 | 38.52 |

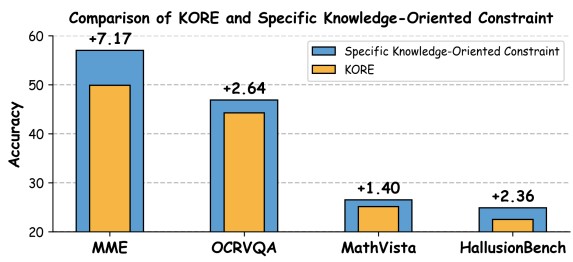

Figure 6: Performance comparison of corresponding tasks under specific knowledge-oriented constraints.

Table 4: Performance comparison between KORE and baseline methods on knowledge adaptation and retention with various LMMs scales and architectures.

| Methods | EVOKE | | COM ↑ | OCR ↑ | M-DIS ↑ | INS ↑ | M-IDU ↑ | MAT ↑ | HAL ↑ | Avg ↑ |
| --- | --- | --- | --- | --- | --- | --- | --- | --- | --- | --- |
| | CEM ↑ | F1 ↑ | | | | | | | | |
| *LLaVA-v1.5 (13B)* | | | | | | | | | | |
| Vanilla | — | — | 66.86 | 51.12 | 52.70 | 66.04 | 33.93 | 19.64 | 56.77 | — |
| LoRA | 16.26 | 22.83 | 60.57 | 32.58 | 43.72 | 23.26 | 17.43 | 15.82 | 38.08 | 25.21 |
| Replay | 12.05 | 20.21 | 65.81 | 47.51 | 48.42 | 61.04 | 24.62 | 19.55 | 54.16 | 30.70 |
| KORE | 32.89 | 44.47 | 59.35 | 45.96 | 51.39 | 65.10 | 26.84 | 20.31 | 40.52 | 41.44 |
| *Qwen2.5-VL (7B)* | | | | | | | | | | |
| Vanilla | — | — | 81.18 | 70.32 | 65.35 | 78.46 | 61.25 | 47.69 | 66.96 | — |
| LoRA | 14.56 | 14.01 | 52.54 | 64.54 | 22.35 | 21.39 | 23.25 | 13.52 | 41.38 | 24.21 |
| Replay | 11.73 | 18.51 | 78.54 | 69.17 | 65.26 | 70.20 | 50.72 | 42.74 | 67.48 | 39.28 |
| KORE | 22.91 | 31.36 | 56.60 | 67.74 | 65.48 | 70.51 | 45.02 | 43.72 | 58.57 | 42.68 |

- **Obs 7: KORE shows enhanced superiority on a larger-scale LMM.** Table 4 shows that KORE surpasses LoRA (*e.g.,* 16.63 ↑ in CEM and 21.64 ↑ in F1-Score) on EVOKE, and achieves superior K.R performance across all six dimensions including M-DIS. With an overall improvement of 10.74 over Replay, these results confirm KORE's strong potential for larger LMMs.
- **Obs 8: KORE 's effectiveness is not architecture-specific.** On Qwen2.5-VL (7B), it surpasses LoRA (*e.g.,* 12.63 ↑ in CEM and 21.27 ↑ in F1-Score) and Replay (*e.g.,* 3.40 ↑ in Avg). Smaller improvement stems from Qwen2.5-VL's robust knowledge system, honed via three-stage training, which reduce marginal gains from knowledge injection (*e.g.,* Comparing Table 1 and 4, Qwen2.5-VL (7B)'s gains are less than LLaVA-v1.5 (7B)'s with LoRA on EVOKE).

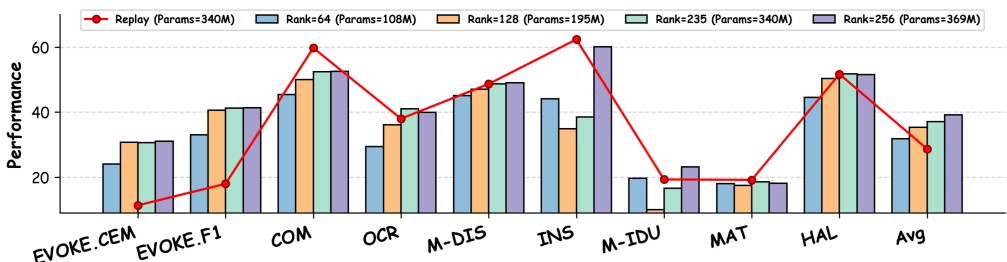

Figure 7: Comparison of different ranks for KORE with LLaVA-v1.5 (7B).

## 4.4 ANALYSIS OF ABLATION EXPERIMENTS

In this section, we conduct extensive ablation studies (*e.g.,* Rank, W/o Augmentation, W/o Constraint and W/o Frozen Matrix $A$) to validate the effectiveness of KORE's design.

Table 5: Comparison of ablation experiment results of KORE on LLaVA-v1.5 (7B).

| Setting | EVOKE | | COM ↑ | OCR ↑ | M-DIS ↑ | INS ↑ | M-IDU ↑ | MAT ↑ | HAL ↑ | Avg ↑ |
|---|---|---|---|---|---|---|---|---|---|---|
| | CEM ↑ | F1 ↑ | | | | | | | | |
| KORE | 30.65 | 41.26 | 52.41 | 40.98 | 48.68 | 38.54 | 16.58 | 18.59 | 51.75 | 37.09 |
| W/o Augmentation | 10.83 | 18.31 | 59.96 | 40.42 | 47.13 | 32.53 | 16.00 | 19.71 | 49.50 | 26.23 |
| W/o Constraint | 33.93 | 43.71 | 46.39 | 32.38 | 46.31 | 32.70 | 15.38 | 19.12 | 46.47 | 36.46 |
| W/o Frozen Matrix $A$ | 31.97 | 41.72 | 50.73 | 39.56 | 48.37 | 35.30 | 16.44 | 19.07 | 49.91 | 36.95 |

- **Obs 9: Larger rank enhance KORE's performance.** Figure 7 shows a clear trend: KORE's performance increases with higher rank and more trainable parameters on nearly all evaluations. KORE (rank=64) still surpasses Replay in Avg, only using less than half of parameters of Replay.
- **Obs 10: Ablation studies reveals the effectiveness of KORE's design.** Table 5 validates KORE's design, showing that each ablated component contributes positively to its overall performance. W/o Augmentation is particularly detrimental to knowledge adaptation (19.82 ↓ in CEM and 22.95 ↓ in F1-Score). Meanwhile, W/o Constraint and W/o Frozen Matrix $A$ impairs knowledge retention.

## 4.5 COMPARISON WITH GENERAL AUGMENTATION METHODS

This section validates our claim from § 3.1 that KORE-AUGMENTATION is superior to general augmentation methods.

- **Obs 11: KORE-AUGMENTATION is superior to general augmentation methods.** In Table 6, KORE-AUGMENTATION outperforms general augmentation methods across all metrics, notably achieving an 18.53 improvement in K.A over the strongest baseline. These results strongly demonstrate that KORE-AUGMENTATION is a highly effective augmentation method.

Table 6: Performance comparison of different augmentation methods.

| Method | K.A ↑ | K.R ↑ | Avg ↑ |
|---|---|---|---|
| KORE-AUGMENTATION | 38.82 | 35.78 | 36.46 |
| *Augmentation for Text* | | | |
| Knowledge-Aware | 20.29 | 34.86 | 27.38 |
| Knowledge-Agnostic | 15.60 | 35.71 | 25.49 |
| *Augmentation for Images* | | | |
| Knowledge-Aware | 18.33 | 34.02 | 25.86 |
| Knowledge-Agnostic | 18.33 | 32.09 | 25.25 |

## 5 LIMITATIONS & FUTURE DISCUSSION

While KORE demonstrates strong performance in knowledge adaptation and retention, we also recognize its limitations. The augmentation process relies on GPT-4o, which may introduce hallucinations, and is confined to enhancing individual knowledge units. Furthermore, extracting covariance matrices from all linear layers is computationally expensive. Future work will explore more structured augmentation (*e.g.,* knowledge graphs and forest (Ji et al., 2021; Chen et al., 2020) with potential for combination with reinforcement learning), and reduce resource consumption by identifying the most critical layers for covariance computation.

## 6 CONCLUSION

In this work, we propose KORE, a synergistic method for knowledge-oriented augmentation and constraint that addresses the critical trade-off between injecting new knowledge and preserving existing knowledge. Specifically, KORE automatically converts each piece of knowledge into a more profound and structured format, ensuring the model accurately learns and adapts to new knowledge. Simultaneously, it minimizes interference with previous knowledge by initializing an adapter with null space that stores the covariance matrix of previous knowledge, enabling powerful retention. KORE's robust performance is architecture-agnostic (*e.g.,* LLaVA-v1.5 and Qwen2.5-VL) and exhibits enhanced superiority on larger-scale LMMs. Furthermore, its capability for specific knowledge-oriented constraints improves retention performance of specific knowledge, granting KORE high flexibility to address diverse scenarios with specialized preservation needs.

### ETHICS STATEMENT

Our KORE method offers significant value for real-world applications in knowledge injection and management by effectively injecting new knowledge while preserving old knowledge. However, while the intention behind knowledge injection is positive, it presents a risk of misuse, such as the introduction of false, harmful, or biased information to compromise the model. We therefore urge the research community to utilize this technology responsibly and cautiously to ensure its ethical application.

### REPRODUCIBILITY STATEMENT

To ensure the reproducibility of our findings, ❶ detailed implementation specifications and hyperparameters for KORE are provided in § B and § H; ❷ all source code will be released upon the completion of the peer-review process; ❸ all training data and weights will be publicly available on Huggingface after the completion of the peer-review process. We hope these resources will enable other researchers in the field to verify and replicate our results.

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

# APPENDIX CONTENTS

## A  THE USE OF LARGE LANGUAGE MODELS IN KORE

In this section, we elaborate on the precise role of large language models within KORE, as detailed below.

- **Usage 1: KORE-74K's construction.** In § 3.1 and § H, we use GPT-4o to generate multi-rounds dialogue data, summary content of original knowledge, and quadruplets $(Q, A, S, H)$ data, which is in line with current scientific research standards
- **Usage 2:Knowledge Retention Evaluation.** In § 4, we employ MIA-Bench, MMDU, MathVista, and MathVision, whose evaluation requires large language models as judges—a practice consistent with current research standards.
- **Usage 3: Paper grammar polishing.** The initial draft of the paper was written by humans and later refined for grammar using large language models, a common practice in contemporary research.

## B  MORE DETAILS ABOUT SETUP AND EXPERIMENTAL OPERATION

### B.1  KNOWLEDGE ADAPTATION EVALUATION

Our knowledge adaptation evaluation completely follows the settings of EVOKE. Below, we will provide an introduction to EVOKE:

**EVOKE:** This paper introduces EVOKE (Jiang et al., 2025), a new benchmark to evaluate how well Large Multimodal Models (LMMs) can learn evolving knowledge without forgetting their original capabilities. It reveals the limitations of current methods in knowledge adaptation and the severity of catastrophic forgetting. The study further shows that knowledge augmentation and continual learning are promising solutions, providing a framework for future research.

### B.2  KNOWLEDGE RETENTION EVALUATION

We evaluate fine-tuned LMMs' knowledge retention capabilities on 12 benchmarks across 7 capability dimensions. And we follow the settings of VLMEvalKit (Duan et al., 2024) to evaluate these benchmarks, and the following is an introduction

1. **MME** (Fu et al., 2023) provides a holistic evaluation of LMMs' perception and cognition across 14 tasks. Its key feature is the use of carefully crafted instruction-answer pairs, which facilitates a straightforward assessment without the need for specialized prompt engineering.
2. **MMBench** (Liu et al., 2024d) is a cross-lingual benchmark for comprehensively evaluating LMMs. It features over 3,000 bilingual multiple-choice questions spanning 20 skill dimensions, from visual recognition to abstract reasoning.
3. **SEEDBench2_Plus** (Li et al., 2024) benchmarks LMMs on interpreting text-rich visuals (*e.g.,* charts, web layouts). It uses 2,300 multiple-choice questions to test reasoning capabilities where integrating textual and visual information is essential.
4. **OCRVQA** (Mishra et al., 2019) is a benchmark for evaluating a model's ability to answer questions by reading text within images. It focuses on tasks where textual information is essential, requiring tight integration of visual perception and OCR.
5. **ScienceQA** (Lu et al., 2022) evaluates scientific reasoning through a large-scale multimodal benchmark; it features curriculum-based questions with diagrams and provides lectures and explanations for each question to encourage complex reasoning.
6. **MMMU** (Yue et al., 2024) evaluates LMMs on college-level, multimodal questions requiring expert knowledge. The benchmark includes 11,500 questions from six disciplines, utilizing 30 image formats to test complex, subject-specific reasoning.
7. **MIA-Bench** (Qian et al., 2024) is a targeted benchmark that measures how precisely LMMs can follow complex and multi-layered instructions. It consists of 400 distinct image-prompt combinations engineered to test a model's ability to comply with detailed and nuanced directives.
8. **MMDU** (Liu et al., 2025b) evaluates LMMs in multi-image, multi-turn conversational scenarios. It specifically assesses a model's capacity for contextual understanding, temporal reasoning, and maintaining coherence throughout extended interactions.

9. **MathVista** (Lu et al., 2024) benchmarks the mathematical reasoning of foundation models in visual contexts. It aggregates 6,141 problems from 31 datasets, requiring detailed visual analysis and compositional logic for solution.

10. **MathVision** (Wang et al., 2025a) provides a challenging dataset of 3,040 visually-presented problems from math competitions. Categorized into 16 mathematical areas and five difficulty tiers, it offers a structured evaluation of advanced reasoning in LMMs.

11. **HallusionBench** (Guan et al., 2024) diagnoses hallucination and illusion in LMMs' visual interpretations. It employs 346 images and 1,129 structured questions to quantitatively analyze the causes of inaccurate or inconsistent model responses.

12. **POPE** (Li et al., 2023) evaluates object hallucination in LMMs—the tendency to describe non-existent objects. It uses a polling-based questioning strategy to reliably measure this tendency.

### B.3 EVALUATION PROTOCOL

To evaluate performance on open-domain question answering tasks, two key metrics are employed: **Cover Exact Match (CEM)** and **F1-Score (F1)**.

The **CEM** metric determines whether the ground truth answer is fully contained within the model's prediction (Xu et al., 2023). It is defined by the equation:

$$CEM = \begin{cases} 1, & y_q \subseteq \hat{Y} \\ 0, & \text{otherwise} \end{cases}$$

where $y_q$ represents the ground truth answer and $\hat{Y}$ is the text generated by the model.

The **F1-Score**, on the other hand, assesses the word-level overlap between the predicted and ground truth answers, providing a harmonic mean of **Precision** and **Recall** (Chan et al., 2024). Given the ground truth as a set of words $\mathcal{W}(y_q) = \{y_1, \ldots, y_m\}$ and the model's prediction as $\mathcal{W}(\hat{Y}) = \{\hat{y}_1, \ldots, \hat{y}_n\}$, the number of common words is calculated as $\mathcal{U}(\hat{Y}, y_q) = \sum_{t \in \mathcal{W}(y_q)} \mathbf{1}[t \in \mathcal{W}(\hat{Y})]$, where $\mathbf{1}[\cdot]$ is the indicator function.

Based on this, **Precision** is the fraction of predicted words that are correct,

$$\mathcal{P}(\hat{Y}, Y) = \frac{\mathcal{U}(\hat{Y}, y_q)}{|\mathcal{W}(\hat{Y})|};$$

while **Recall** is the fraction of ground truth words that were successfully predicted,

$$\mathcal{R}(\hat{Y}, Y) = \frac{\mathcal{U}(\hat{Y}, y_q)}{|\mathcal{W}(Y)|}.$$

### B.4 BASELINE METHODS

In this section, we provide a brief introduction to the baseline method, as follows:

**EWC:** This seminal continual learning work (Kirkpatrick et al., 2017) introduces Elastic Weight Consolidation (EWC) to mitigate catastrophic forgetting. EWC slows updates to parameters important for prior tasks by imposing a quadratic constraint based on the Fisher Information Matrix, elastically preserving old knowledge during new learning.

**LwF:** This work proposes Knowledge Distillation to mitigate catastrophic forgetting (Li & Hoiem, 2017b). The method preserves knowledge by ensuring the new model's predictions on new data align with the old model's outputs, achieving data-free continual learning through output consistency.

**LoRA:** This highly efficient method, **LoRA** (Hu et al., 2022), fine-tunes models by training only small, injected low-rank matrices while keeping the original weights frozen. This approach reduces computational costs significantly and helps mitigate catastrophic forgetting.

**OLoRA:** This work proposes an orthogonal subspace-based method for continual learning (Wang et al., 2023). It allocates independent, orthogonal parameter subspaces for each task, constraining updates to prevent interference and mitigate catastrophic forgetting via an elegant geometric solution.

**MoELoRA:** The method in (Luo et al., 2024) combines MoE with contrastive learning for PEFT. It specializes experts for different data types and uses contrastive objectives to guide expert collaboration, achieving parameter-efficient fine-tuning that reduces catastrophic forgetting.

**SEFE:** The method in (Chen et al., 2025) tackles multimodal catastrophic forgetting by separately addressing two types: superficial forgetting of style and essential forgetting of knowledge. A tailored training strategy preserves essential knowledge during continual instruction learning.

### B.5 TRAINING PARAMETERS ABOUT KORE

We have displayed some training parameter settings, as shown in Table 7.

Table 7: Hyperparameter settings for the model training on LLaVA-v1.5 (7B), LLaVA-v1.5 (13B) and Qwen2.5-VL (7B).

| *LLaVA-v1.5 (7B)* | | | | |
|---|---|---|---|---|
| **Rank** | **Optimizer** | **Deepspeed** | **Epochs** | **Vision Select Layer** |
| 235 | AdamW | Zero3 | 6 | -2 |
| **Weight Decay** | **Warmup Ratio** | **LR Schedule** | **Learning Rate** | **Batch Size** |
| 0 | 0.03 | cosine decay | $2 \times 10^{-4}$ | 54 |
| *LLaVA-v1.5 (13B)* | | | | |
| **Rank** | **Optimizer** | **Deepspeed** | **Epochs** | **Vision Select Layer** |
| 235 | AdamW | Zero3 | 6 | -2 |
| **Weight Decay** | **Warmup Ratio** | **LR Schedule** | **Learning Rate** | **Batch Size** |
| 0 | 0.03 | cosine decay | $2 \times 10^{-4}$ | 32 |
| *Qwen2.5-VL (7B)* | | | | |
| **Rank** | **Optimizer** | **Deepspeed** | **Epochs** | **Image Max Pixels** |
| 274 | AdamW | Zero3 | 6 | 262144 |
| **Grad Accum Steps** | **Warmup Ratio** | **LR Schedule** | **Learning Rate** | **Batch Size** |
| 8 | 0.1 | cosine decay | $2 \times 10^{-4}$ | 24 |

### B.6 EXPERIMENT RESOURCES ABOUT KORE

All training experiments were conducted using 4 NVIDIA H100 GPUs (each with 96 GiB memory). All evaluation experiments were performed on systems equipped with 4 NVIDIA A100 PCIe GPUs (each with 40 GiB memory).

## C PROOF OF KORE

In Section § 3.2, KORE-Constraint initializes the LoRA's low-rank matrix $A$ within the null space of the covariance matrix $C$, which represents prior knowledge and capabilities. This claim is the premise for KORE-Constraint's effectiveness, which we formally prove in Theorem 1.

In KORE, the LoRA's low-rank matrix $A$ is frozen, and only matrix $B$ is fine-tuned during the process. We demonstrate in Theorem 2 why this operation minimizes interference with prior knowledge and capabilities during fine-tuning. Theorem 2 extends Theorem 1: as long as Theorem 1 ensures that matrix $A$ lies in the null space of the covariance matrix $C$, the final output of each layer, $W^*X$, remains approximately equal to $W_0X$, regardless of how the parameters of matrix $B$ are adjusted.

**Theorem 1.** *Let $U_{null}^T$, $W_0$, $A$ be the approximate null space of the model's covariance matrix composed of input activations in linear layers, the pre-training weights of the model, and the low rank matrix in LoRA, respectively.*

*Proof.* We aim to prove that under the assumption that $W_0$ is full-rank, the column space of $A$ forms a subset of the column space of $U_{null}^T$, which means $\text{Col}(A) = \text{Col}(U_{null}^T)$.

Step 1: Based on the definition in Section § 3.2:

$$A = \sqrt{\Sigma^*}(V^*)^T \tag{4}$$

Since $\Sigma^*$ is a diagonal matrix containing singular values, it only scales the columns of $(V^*)^T$ without changing their span. Therefore, the column space of matrix $A$ is identical to the column space of $(V^*)^T$:

$$\text{Col}(A) = \text{Col}((V^*)^T) \tag{5}$$

Step 2: Based on the SVD of $W_0 U_{\text{null}} U_{\text{null}}^T$ in Section § 3.2:

$$W_0 U_{\text{null}} U_{\text{null}}^T = U^* \Sigma^* (V^*)^T \tag{6}$$

$V^*$ represents the right singular vectors of $W_0 U_{\text{null}} U_{\text{null}}^T$ and spans its row space. Since $U_{\text{null}}$ is orthogonal, $U_{\text{null}} U_{\text{null}}^T$ projects any matrix onto the subspace spanned by $U_{\text{null}}$. Therefore, when $W_0$ is full-rank, the column space of $W_0 U_{\text{null}} U_{\text{null}}^T$ is identical to the column space of $U_{\text{null}}^T$:

$$\text{Col}(V^*) = \text{Col}(W_0 U_{\text{null}} U_{\text{null}}^T) = \text{Col}(U_{\text{null}}^T) \tag{7}$$

Step 3: Combining the content of steps 1 and 2:

$$\text{Col}(A) = \text{Col}(V^*) = \text{Col}(U_{\text{null}}^T) \tag{8}$$

Thus, the column space of $A$ is identical to the column space of $U_{\text{null}}^T$, completing the proof.

**Theorem 2.** *Let $X^{(l)}$, $W_0^{(l)}$, and $W^{*(l)}$ denote the input activations from pre-trained knowledge, the initial weight matrix of the $l$-th layer before fine-tuning, and the weight matrix of the $l$-th layer after fine-tuning, respectively, for the $l$-th layer of the LMM.*

*Proof.* We aim to prove that the output of the $l$-th layer remains approximately unchanged after fine-tuning with KORE, *e.g.,*

$$W^{*(l)} X^{(l)} \approx W_0^{(l)} X^{(l)}, \tag{9}$$

In KORE, the fine-tuned output at the $l$-th layer is defined as:

$$W^{*(l)} = W_0^{(l)} - B^{(l)} A^{(l)} + B^{*(l)} A^{(l)}. \tag{10}$$

Based on $A^{(l)} X^{(l)} \approx 0$ from Section § 3.2, we have:

$$W^{*(l)} X^{(l)} \approx W_0^{(l)} X^{(l)}. \tag{11}$$

The output thus remains approximately unchanged, ensuring that the fine-tuning minimally alters the pre-trained knowledge. This concludes the proof.

## D    MORE DETAILS ABOUT ANALYSIS OF ABILITY TO CAPTURE KNOWLEDGE

### D.1    DETAILED EXPERIMENTAL RESULTS FOR CAPTURE KNOWLEDGE

Table 8 presents detailed data and additional results from the experiment illustrated in Figure 4. The results indicate that the number of sampled data points has only a limited influence. When the smallest 1536 ranks are discarded, performance with 512 samples is slightly lower than with 256 samples; using 32 samples leads to a more noticeable decline compared to 256 samples, yet still significantly outperforms both Plain SVD and ASVD (Yuan et al., 2023). This suggests that even a small number of samples is sufficient to capture essential knowledge into the covariance matrix.

Furthermore, using test-specific samples allows for better performance after discarding a large number of ranks. For instance, when discarding 1536 ranks, CO-SVD (with 256 MME samples) outperforms CO-SVD (with 256 ScienceQA samples) on the MME, while CO-SVD (with 256 ScienceQA samples) surpasses CO-SVD (with 256 MME samples) on ScienceQA. This demonstrates that CO-SVD effectively captures dataset-specific knowledge and preserves structural features in the covariance matrix, enabling knowledge-oriented constraints and resulting in powerful retention.

Table 8: The detailed numbers and more results of the experiment in Figure 4

| Test Data | Method | Discarded Ranks | | | | |
|---|---|---|---|---|---|---|
| | | 128 | 256 | 512 | 1024 | 1536 |
| MME | Plain SVD | 1492.95 | 1487.28 | 1318.18 | 1169.87 | 744.03 |
| | ASVD (with 256 MME samples) | 1490.14 | 1476.02 | 1488.48 | 1425.41 | 1239.74 |
| | CO-SVD (with 256 MME samples) | 1498.17 | 1511.25 | 1514.43 | 1486.81 | **1458.36** |
| | CO-SVD (with 32 MME samples) | 1508.90 | 1512.90 | 1507.78 | 1498.81 | 1341.82 |
| | CO-SVD (with 512 MME samples) | 1507.42 | 1516.68 | 1505.33 | 1460.32 | 1449.82 |
| | CO-SVD (with 256 ScienceQA samples) | 1486.51 | 1492.65 | 1478.73 | 1419.61 | 1300.89 |
| ScienceQA | Plain SVD | 67.13 | 66.85 | 65.59 | 50.41 | 0.73 |
| | ASVD (with 256 ScienceQA samples) | 67.63 | 66.95 | 66.75 | 62.38 | 49.14 |
| | CO-SVD (with 256 ScienceQA samples) | 67.19 | 67.16 | 67.62 | 67.61 | **66.76** |
| | CO-SVD (with 32 ScienceQA samples) | 67.48 | 66.77 | 66.97 | 66.61 | 64.58 |
| | CO-SVD (with 512 ScienceQA samples) | 67.08 | 67.00 | 67.40 | 66.91 | 66.27 |
| | CO-SVD (with 256 MME samples) | 67.74 | 67.49 | 67.53 | 65.69 | 62.43 |

### D.2    COVARIANCE VISUALIZATION RESULTS

In Figures 8 and 9, we further provide visualizations of the covariance matrices collected from the POPE, HallusionBench, and MMBench tasks.

Due to the high and uninformative original dimensionality of 4096 or 11088, we downsampled the covariance matrices to 32×32 and visualized their heatmaps. We present activations prior to various linear weights, including "mlp.down_proj", "mlp.gate_proj","self_attn.v_proj"and "self_attn.o_proj" from both **layer 2** and **layer 30**. The results show that heatmaps from **POPE** and **Hallusion-Bench**—both hallucination evaluation tasks—share certain similar patterns (highlighted with red circles) not observed in heatmaps from **MMBench**. This indicates that the activated covariance matrices exhibit distinct patterns when inputs from different tasks are processed by the LMMs. These visualizations empirically support that covariance matrix patterns can characterize the triggered task. We leverage such patterns to guide the decomposition of pre-trained weights in LMMs, obtaining initialized adapters enriched with more informative knowledge.

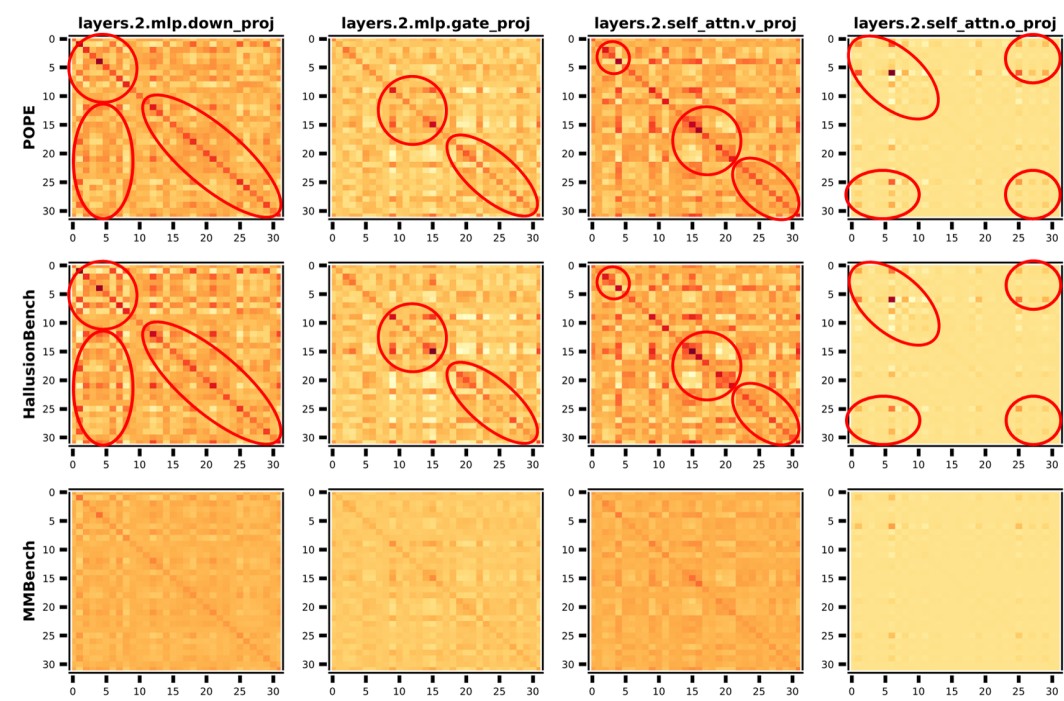

Figure 8: Covariance matrix visualization for "mlp.down_proj", "mlp.gate_proj", "self_attn.v_proj" and "self_attn.o_proj" weights in the **2-th layer** on **POPE, HallusionBench and MMBench**.

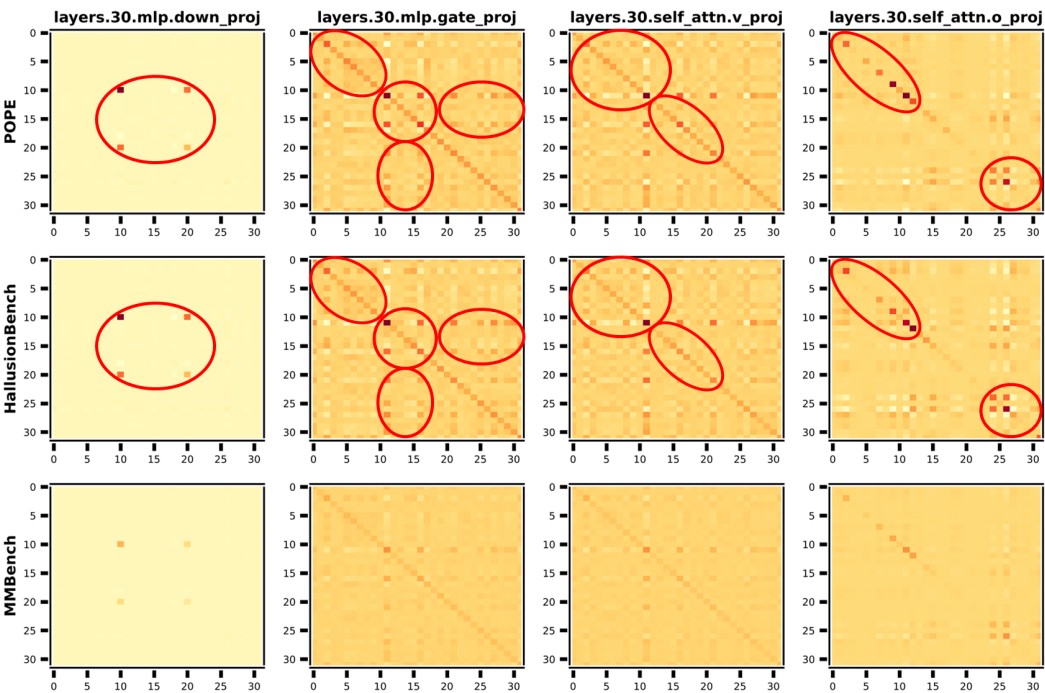

Figure 9: Covariance matrix visualization for "mlp.down_proj", "mlp.gate_proj", "self_attn.v_proj" and "self_attn.o_proj" weights in the **30-th layer** on **POPE, HallusionBench and MMBench**.

# E    MORE EXPERIMENTAL RESULTS ABOUT KORE

## E.1    MORE MAIN RESULTS

Regarding the experiment in Figure 5 in § 4.2, we have supplemented Table 9 with detailed numerical performance of all methods on fine-grained knowledge types for readers' reference.

Table 9: **Performance comparison between KORE and baseline methods on fine-grained knowledge types with LLaVA-v1.5 (7B).** PO: Politics; SP: Sports; BU: Business; HE: Health; CE: Celebrity; FI: Film; AL: Album; WR: Written Work.

| Method | News Avg | | PO | | SP | | BU | | HE | | Entity Avg | | CE | | FI | | AL | | WR | |
|---|---|---|---|---|---|---|---|---|---|---|---|---|---|---|---|---|---|---|---|---|
| | CEM↑ | F1↑ | CEM↑ | F1↑ | CEM↑ | F1↑ | CEM↑ | F1↑ | CEM↑ | F1↑ | CEM↑ | F1↑ | CEM↑ | F1↑ | CEM↑ | F1↑ | CEM↑ | F1↑ | CEM↑ | F1↑ |
| Full-FT | 21.35 | 16.34 | 12.92 | 10.99 | 22.49 | 20.88 | 27.31 | 20.95 | 19.84 | 16.47 | 14.37 | 13.88 | 13.11 | 16.93 | 12.39 | 13.16 | 12.17 | 7.66 | 20.34 | 8.43 |
| LoRA | 17.72 | 19.42 | 10.54 | 12.96 | 19.11 | 21.50 | 20.66 | 24.03 | 17.81 | 23.76 | 12.51 | 17.09 | 12.20 | 21.19 | 10.57 | 15.82 | 10.72 | 8.72 | 18.64 | 12.94 |
| Replay | 13.98 | 19.43 | 7.61 | 13.16 | 15.96 | 20.69 | 16.05 | 22.40 | 15.38 | 24.21 | 8.48 | 16.39 | 9.40 | 18.78 | 10.34 | 15.60 | 3.77 | 10.79 | 4.55 | 8.23 |
| EWC | 17.86 | 21.10 | 10.45 | 14.81 | 19.83 | 23.02 | 19.00 | 24.57 | 17.41 | 23.88 | 12.88 | 17.58 | 14.53 | 22.07 | 12.16 | 16.91 | 10.72 | 8.13 | 15.25 | 17.69 |
| LwF | 17.05 | 21.43 | 9.62 | 13.99 | 19.83 | 23.66 | 18.63 | 25.82 | 19.03 | 26.20 | 11.88 | 18.40 | 12.45 | 21.64 | 12.39 | 17.01 | 9.28 | 11.11 | 10.17 | 17.10 |
| MoELoRA | 9.23 | 14.86 | 3.39 | 8.72 | 6.77 | 11.77 | 12.36 | 18.92 | 10.53 | 20.60 | 3.40 | 9.28 | 2.95 | 10.32 | 4.43 | 8.96 | 3.19 | 5.22 | 10.17 | 14.07 |
| O-LoRA | 9.21 | 14.68 | 3.67 | 8.52 | 7.01 | 12.23 | 12.55 | 18.98 | 11.74 | 20.68 | 3.40 | 9.22 | 3.10 | 10.51 | 4.20 | 8.28 | 3.19 | 5.35 | 8.47 | 12.37 |
| SEFE | 16.66 | 18.44 | 10.82 | 12.64 | 17.78 | 20.92 | 20.30 | 23.23 | 17.00 | 21.55 | 9.79 | 15.18 | 10.77 | 20.13 | 9.09 | 12.01 | 5.51 | 7.47 | 13.56 | 13.87 |
| KORE | 34.74 | 42.96 | 23.83 | 32.31 | 46.19 | 50.38 | 34.69 | 45.74 | 33.20 | 45.23 | 26.17 | 39.39 | 27.79 | 42.61 | 26.93 | 34.05 | 16.52 | 29.54 | 28.81 | 43.05 |

## E.2 MORE RESULTS ON LMM SCALES AND ARCHITECTURES

Regarding the experiment in § 4.3, we have supplemented the detailed results of knowledge adaptation and retention in Tables 10 and 11, respectively.

Table 10: Performance comparison between KORE and baseline methods on fine-grained knowledge retention evaluations with LLaVA-v1.5 (13B) and Qwen2.5-VL (7B).

| Method | COM | | OCR | | M-DIS | | INS | M-IDU | MAT | | HAL | | Avg |
|---|---|---|---|---|---|---|---|---|---|---|---|---|---|
| | MME↑ | MM$^B$↑ | SEED$^{B2P}$↑ | OCR$^{VQA}$↑ | SQA↑ | MMMU↑ | MIA$^B$↑ | MMDU↑ | Math$^T$↑ | Math$^I$↑ | POPE↑ | Hall$^B$↑ | |
| | | | | | | *LLaVA-v1.5 (13B)* | | | | | | | |
| Vanilla | 65.33 | 68.38 | 42.25 | 59.99 | 73.90 | 31.50 | 66.04 | 33.93 | 27.40 | 11.88 | 87.07 | 26.46 | 49.51 |
| LoRA | 30.00 | 60.57 | 36.93 | 28.22 | 69.13 | 18.30 | 23.26 | 17.43 | 23.90 | 7.73 | 71.64 | 4.52 | 32.64 |
| Replay | 57.49 | 65.81 | 40.27 | 54.75 | 70.94 | 25.90 | 61.04 | 24.62 | 27.00 | 12.11 | 87.09 | 21.23 | 45.69 |
| KORE | 55.99 | 62.71 | 40.32 | 51.60 | 71.97 | 30.80 | 65.10 | 26.84 | 27.30 | 13.32 | 79.29 | 18.91 | 45.35 |
| | | | | | | *Qwen2.5-VL (7B)* | | | | | | | |
| Vanilla | 82.54 | 79.81 | 69.61 | 71.03 | 72.10 | 58.60 | 78.46 | 61.25 | 69.70 | 25.69 | 86.51 | 47.42 | 66.89 |
| LoRA | 67.88 | 37.20 | 59.29 | 69.79 | 42.30 | 2.40 | 21.39 | 23.25 | 39.40 | 13.52 | 73.73 | 9.02 | 38.16 |
| Replay | 75.38 | 81.70 | 69.16 | 69.17 | 85.12 | 45.40 | 70.20 | 50.72 | 63.90 | 21.58 | 87.49 | 47.48 | 63.94 |
| KORE | 36.23 | 76.98 | 66.80 | 68.69 | 85.55 | 45.40 | 70.51 | 45.02 | 63.10 | 24.34 | 75.24 | 41.89 | 58.31 |

- **Obs 1 in § E.2: KORE still achieves superior knowledge retention performance on larger-scale LMM and different model architectures.** As shown in Table 10, on LLaVA-v1.5 (13B), KORE outperforms Replay on seven benchmarks and achieves comparable overall performance. This result demonstrates KORE's potential for superior performance on larger-scale LMM. On Qwen2.5-VL (7B), KORE surpasses LoRA by 20.15 in overall performance, demonstrating its ability to maintain superior knowledge retention across different model architectures and confirming its universality and robustness.

Table 11: Performance comparison between KORE and baseline methods on fine-grained knowledge types with LLaVA-v1.5 (13B) and Qwen2.5-VL (7B).

| Method | News Avg | | PO | | SP | | BU | | HE | | Entity Avg | | CE | | FI | | AL | | WR | |
|---|---|---|---|---|---|---|---|---|---|---|---|---|---|---|---|---|---|---|---|---|
| | CEM↑ | F1↑ | CEM↑ | F1↑ | CEM↑ | F1↑ | CEM↑ | F1↑ | CEM↑ | F1↑ | CEM↑ | F1↑ | CEM↑ | F1↑ | CEM↑ | F1↑ | CEM↑ | F1↑ | CEM↑ | F1↑ |
| | | | | | | | | *LLaVA-v1.5 (13B)* | | | | | | | | | | | | |
| LoRA | 20.15 | 25.10 | 12.65 | 16.17 | 24.79 | 28.69 | 21.77 | 29.51 | 22.27 | 29.09 | 11.99 | 20.34 | 13.72 | 25.26 | 13.18 | 18.04 | 6.67 | 12.18 | 10.17 | 15.87 |
| Replay | 15.04 | 21.83 | 8.16 | 14.41 | 15.60 | 21.76 | 15.87 | 24.74 | 18.62 | 28.74 | 8.77 | 18.42 | 9.45 | 21.50 | 10.91 | 17.16 | 5.51 | 13.38 | 10.17 | 20.97 |
| KORE | 36.77 | 46.11 | 25.39 | 34.41 | 47.16 | 53.39 | 37.45 | 50.95 | 35.22 | 48.51 | 28.64 | 42.67 | 28.66 | 44.95 | 31.02 | 38.21 | 22.61 | 35.43 | 20.34 | 33.06 |
| | | | | | | | | *Qwen2.5-VL (7B)* | | | | | | | | | | | | |
| LoRA | 17.76 | 14.09 | 12.01 | 7.18 | 17.41 | 17.65 | 22.32 | 17.90 | 19.03 | 17.21 | 11.06 | 13.93 | 8.03 | 15.91 | 21.48 | 14.91 | 8.70 | 10.87 | 16.95 | 11.32 |
| Replay | 13.45 | 18.40 | 7.33 | 11.09 | 14.03 | 17.94 | 14.58 | 22.72 | 15.38 | 23.72 | 9.84 | 18.63 | 7.16 | 17.69 | 20.45 | 28.00 | 9.28 | 12.97 | 16.95 | 24.89 |
| KORE | 26.93 | 32.51 | 17.42 | 22.75 | 31.20 | 35.11 | 31.00 | 39.43 | 33.20 | 40.49 | 18.51 | 30.11 | 16.11 | 28.63 | 26.14 | 33.20 | 13.33 | 25.91 | 25.42 | 41.24 |

- **Obs 2 in § E.2: KORE achieves comprehensive performance advantages across diverse knowledge types on both larger-scale LMM and different model architectures.** In Table 11, KORE achieves the best knowledge adaptation performance across all news and entity types on both LLaVA-v1.5 (13B) and Qwen2.5-VL (7B), significantly outperforming LoRA and Replay. This demonstrates that KORE's effectiveness in new knowledge injection is not constrained by model scale or architecture, highlighting its powerful universality.

### E.3 MORE RESULTS ON SPECIFIC KNOWLEDGE-ORIENTED CONSTRAIN

For the experiment on specific knowledge-oriented constraints in § 4.2, we have provided detailed results and presented them below.

Table 12: Performance of specific knowledge-oriented constrains in knowledge adaptation and retention with LLaVA-v1.5 (7B).

| Methods | EVOKE CEM ↑ | EVOKE F1 ↑ | COM ↑ | OCR ↑ | M-DIS ↑ | INS ↑ | M-IDU ↑ | MAT ↑ | HAL ↑ | Avg ↑ |
|---|---|---|---|---|---|---|---|---|---|---|
| KORE | **30.65** | **41.26** | 52.41 | 40.98 | 48.68 | 38.54 | 16.58 | 18.59 | 51.75 | 37.09 |
| KORE$_{MME}$ | 29.48 | 39.44 | **56.90** | 39.86 | 47.41 | 60.10 | **27.70** | 17.92 | 52.20 | 38.81 |
| KORE$_{OCR^{VQA}}$ | 29.95 | 39.75 | 52.60 | **41.47** | **48.86** | 57.06 | 27.09 | 18.28 | 50.15 | 38.53 |
| KORE$_{Math^T}$ | 30.06 | 40.33 | 52.40 | 40.32 | 48.57 | 60.30 | 27.69 | **19.24** | 51.57 | **39.03** |
| KORE$_{Hall^B}$ | 29.93 | 39.98 | 54.37 | 36.68 | 46.50 | **60.71** | 26.30 | 17.42 | **52.67** | 38.52 |

- **Obs 1 in § E.3: KORE with specific knowledge-oriented constraints achieves superior comprehensive performance.** In Table 12, KORE with specific knowledge-oriented constraints (*e.g.*, MME, OCR$^{VQA}$, Math$^T$, Hall$^B$) causes a slight decrease in knowledge adaptation efficacy, it yields a significant increase in knowledge retention performance on INS and M-IDU, resulting in a superior overall performance.
- **Obs 2 in § E.3: Specific knowledge-oriented constraints enhance the retention of corresponding knowledge.** In Table 13, specific knowledge-oriented constraints enhance the retention of corresponding knowledge without compromising the retention of other knowledge types. This capability underscores KORE's potential for applications requiring customized knowledge preservation.

Table 13: Performance of specific knowledge-oriented constrains on fine-grained knowledge retention evaluations with LLaVA-v1.5 (7B).

| Method | COM MME ↑ | COM MM$^B$ ↑ | OCR SEED$^{B2P}$ ↑ | OCR OCR$^{VQA}$ ↑ | M-DIS SQA ↑ | M-DIS MMMU$^T$ ↑ | INS MIA$^B$ ↑ | M-IDU MMDU ↑ | MAT Math$^T$ ↑ | MAT Math$^I$ ↑ | HAL POPE ↑ | HAL Hall$^B$ ↑ | Avg |
|---|---|---|---|---|---|---|---|---|---|---|---|---|---|
| KORE | 49.84 | 54.98 | **37.73** | 44.24 | 68.06 | 29.30 | 38.54 | 16.58 | 25.10 | 12.09 | 80.99 | 22.51 | 40.00 |
| KORE$_{MME}$ | **57.01** | **56.79** | 37.51 | 42.22 | 66.83 | 28.00 | 60.10 | **27.70** | 24.00 | 11.84 | **81.62** | 22.79 | **43.03** |
| KORE$_{OCR^{VQA}}$ | 50.81 | 54.38 | 36.06 | **46.88** | **68.22** | 29.50 | 57.06 | 27.09 | 24.30 | **12.27** | 80.82 | 19.47 | 42.24 |
| KORE$_{Math^T}$ | 48.87 | 55.93 | 36.41 | 44.24 | 67.23 | **29.90** | 60.30 | 27.69 | **26.50** | 11.97 | 81.04 | 22.09 | 42.68 |
| KORE$_{Hall^B}$ | 55.31 | 53.44 | 35.18 | 38.18 | 67.30 | 25.70 | **60.71** | 26.30 | 23.10 | 11.74 | 80.46 | **24.87** | 41.86 |

- **Obs 3 in § E.3: Specific knowledge-oriented constraints also achieve excellent adaptation performance across a wide spectrum of fine-grained knowledge.** In Table 14, KORE with specific knowledge-oriented constraints maintains strong adaptation performance across various News and Entity knowledge types, with negligible performance degradation.

Table 14: Performance of specific knowledge-oriented constrains on fine-grained knowledge types with LLaVA-v1.5 (7B).

| Method | News Avg CEM ↑ | News Avg F1 ↑ | PO CEM ↑ | PO F1 ↑ | SP CEM ↑ | SP F1 ↑ | BU CEM ↑ | BU F1 ↑ | HE CEM ↑ | HE F1 ↑ | Entity Avg CEM ↑ | Entity Avg F1 ↑ | CE CEM ↑ | CE F1 ↑ | FI CEM ↑ | FI F1 ↑ | AL CEM ↑ | AL F1 ↑ | WR CEM ↑ | WR F1 ↑ |
|---|---|---|---|---|---|---|---|---|---|---|---|---|---|---|---|---|---|---|---|---|
| KORE | 34.74 | 42.96 | 23.83 | 32.31 | 46.19 | 50.38 | 34.69 | 45.74 | 33.20 | 45.23 | 26.17 | 39.39 | 27.79 | 42.61 | 26.93 | 34.05 | 16.52 | 29.54 | 28.81 | 43.05 |
| KORE$_{MME}$ | 34.05 | 41.53 | 23.92 | 31.46 | 43.17 | 47.28 | 34.32 | 46.12 | 35.63 | 45.38 | 24.48 | 37.15 | 27.24 | 40.96 | 22.61 | 30.43 | 15.07 | 27.72 | 30.51 | 42.16 |
| KORE$_{OCR^{VQA}}$ | 34.46 | 41.66 | 24.29 | 31.69 | 43.53 | 48.34 | 36.35 | 46.09 | 33.20 | 44.35 | 25.01 | 37.65 | 27.24 | 41.17 | 24.09 | 31.60 | 14.78 | 27.16 | 30.51 | 42.17 |
| KORE$_{Math^T}$ | 33.71 | 41.72 | 22.27 | 30.39 | 45.95 | 50.88 | 33.03 | 43.38 | 30.77 | 43.55 | 26.06 | 38.82 | 28.15 | 42.46 | 25.80 | 32.97 | 15.07 | 27.37 | 30.51 | 42.11 |
| KORE$_{Hall^B}$ | 34.23 | 41.74 | 24.11 | 32.09 | 43.05 | 46.98 | 35.06 | 44.92 | 32.39 | 43.53 | 25.21 | 38.05 | 27.54 | 41.68 | 24.66 | 32.34 | 14.78 | 26.86 | 28.81 | 40.13 |

### E.4 MORE RESULTS ON ABLATION EXPERIMENTS

Regarding the experiment in § 4.4, we have supplemented the experiments in § E.4.1 and § E.4.2.

#### E.4.1 RANK ABLATION EXPERIMENTS

- **Obs 1 in § E.4.1: Increasing the number of trainable parameters enables KORE to achieve stronger performance.** In Table 15, KORE's performance in both knowledge adaptation and knowledge retention exhibits a consistent upward trend as the rank and number of trainable

Table 15: Performance comparison across different ranks in knowledge adaptation and retention with LLaVA-v1.5 (7B).

| Methods | EVOKE | | COM ↑ | OCR ↑ | M-DIS ↑ | INS ↑ | M-IDU ↑ | MAT ↑ | HAL ↑ | Avg ↑ |
| --- | --- | --- | --- | --- | --- | --- | --- | --- | --- | --- |
| | CEM ↑ | F1 ↑ | | | | | | | | |
| KORE (rank=64) | 24.00 | 33.07 | 45.35 | 29.46 | 45.02 | 44.07 | 19.62 | 18.08 | 44.48 | 31.81 |
| KORE (rank=128) | 30.72 | 40.55 | 49.97 | 36.05 | 47.07 | 34.87 | 10.00 | 17.46 | 50.30 | 35.37 |
| KORE (rank=235) | 30.65 | 41.26 | 52.41 | 40.98 | 48.68 | 38.54 | 16.58 | 18.59 | 51.75 | 37.09 |
| KORE (rank=256) | 31.05 | 41.32 | 52.48 | 39.96 | 48.96 | 60.02 | 23.18 | 18.09 | 51.50 | 39.11 |

parameters increase. This trend is particularly significant on the INS and M-IDU dimensions, which indicates KORE's potential to achieve even stronger performance with larger parameter.

Table 16: Performance of comparison across different ranks on fine-grained knowledge retention evaluations with LLaVA-v1.5 (7B).

| Method | COM | | OCR | | M-DIS | | INS | M-IDU | MAT | | HAL | | Avg |
| --- | --- | --- | --- | --- | --- | --- | --- | --- | --- | --- | --- | --- | --- |
| | MME ↑ | MM$^B$ ↑ | SEED$^{B2P}$ ↑ | OCR$^{VQA}$ ↑ | SQA ↑ | MMMU$^T$ ↑ | MIA$^B$ ↑ | MMDU ↑ | Math$^T$ ↑ | Math$^I$ ↑ | POPE ↑ | Hall$^B$ ↑ | |
| KORE (rank=64) | 43.63 | 47.08 | 33.55 | 25.36 | 66.34 | 23.70 | 44.07 | 19.62 | 25.20 | 10.95 | 74.22 | 14.73 | 35.70 |
| KORE (rank=128) | 47.96 | 51.98 | 36.32 | 35.77 | 67.44 | 26.70 | 34.87 | 10.00 | 23.90 | 11.02 | 79.63 | 20.97 | 37.21 |
| KORE (rank=235) | 49.84 | 54.98 | 37.73 | 44.24 | 68.06 | 29.30 | 38.54 | 16.58 | 25.10 | 12.09 | 80.99 | 22.51 | 40.00 |
| KORE (rank=256) | 50.06 | 54.90 | 36.89 | 43.03 | 68.51 | 29.40 | 60.02 | 23.18 | 24.70 | 11.48 | 80.77 | 22.23 | 42.10 |

- **Obs 2 in § E.4.1: Larger trainable parameter scales enhance KORE's knowledge retention performance.** In Table 16, KORE (rank=256) achieves near-comprehensive superiority across 12 benchmarks and surpasses KORE (rank=235) by 2.10 in overall performance. This underscores that a larger trainable parameter scale activates stronger knowledge retention in KORE.

Table 17: Performance comparison across different ranks on fine-grained knowledge types with LLaVA-v1.5 (7B).

| Method | News | | | | | | | | | | Entity | | | | | | | | | |
| --- | --- | --- | --- | --- | --- | --- | --- | --- | --- | --- | --- | --- | --- | --- | --- | --- | --- | --- | --- | --- |
| | Avg | | PO | | SP | | BU | | HE | | Avg | | CE | | FI | | AL | | WR | |
| | CEM ↑ | F1 ↑ | CEM ↑ | F1 ↑ | CEM ↑ | F1 ↑ | CEM ↑ | F1 ↑ | CEM ↑ | F1 ↑ | CEM ↑ | F1 ↑ | CEM ↑ | F1 ↑ | CEM ↑ | F1 ↑ | CEM ↑ | F1 ↑ | CEM ↑ | F1 ↑ |
| KORE (rank=64) | 28.31 | 34.84 | 20.44 | 27.66 | 36.64 | 41.11 | 28.60 | 38.13 | 26.72 | 35.77 | 19.27 | 31.11 | 21.24 | 35.25 | 18.98 | 25.33 | 11.01 | 23.14 | 22.03 | 33.44 |
| KORE (rank=128) | 34.70 | 42.07 | 24.20 | 31.56 | 44.50 | 49.17 | 36.72 | 47.68 | 34.82 | 44.39 | 26.35 | 38.89 | 28.81 | 43.19 | 23.86 | 30.22 | 17.97 | 28.77 | 35.59 | 44.86 |
| KORE (rank=235) | 34.74 | 42.96 | 23.83 | 32.31 | 46.19 | 50.38 | 34.69 | 45.74 | 33.20 | 45.23 | 26.17 | 39.39 | 27.79 | 42.61 | 26.93 | 34.05 | 16.52 | 29.54 | 28.81 | 43.05 |
| KORE (rank=256) | 35.17 | 42.98 | 23.92 | 31.24 | 45.83 | 50.35 | 35.98 | 47.11 | 32.79 | 43.80 | 26.55 | 39.49 | 28.46 | 42.74 | 27.16 | 34.52 | 15.65 | 26.81 | 27.12 | 39.92 |

- **Obs 3 in § E.4.1: Larger trainable parameters improve KORE's knowledge adaptation performance on News and Entity types.** In Table 17, KORE (rank=256) achieves robust and consistent performance across a broader range of fine-grained knowledge types, demonstrating KORE's potential for superior performance with an increased number of trainable parameters.

E.4.2 SETTING ABLATION EXPERIMENTS

Table 18: Performance comparison of setting ablation in knowledge retention with LLaVA-v1.5 (7B).

| Method | COM | | OCR | | M-DIS | | INS | M-IDU | MAT | | HAL | | Avg |
| --- | --- | --- | --- | --- | --- | --- | --- | --- | --- | --- | --- | --- | --- |
| | MME ↑ | MM$^B$ ↑ | SEED$^{B2P}$ ↑ | OCR$^{VQA}$ ↑ | SQA ↑ | MMMU$^T$ ↑ | MIA$^B$ ↑ | MMDU ↑ | Math$^T$ ↑ | Math$^I$ ↑ | POPE ↑ | Hall$^B$ ↑ | |
| KORE | 49.84 | 54.98 | 37.73 | 44.24 | 68.06 | 29.30 | 38.54 | 16.58 | 25.10 | 12.09 | 80.99 | 22.51 | 51.75 |
| W/o Augmentation | 58.75 | 61.17 | 36.80 | 44.04 | 68.15 | 26.10 | 32.53 | 16.00 | 28.00 | 11.41 | 81.29 | 17.71 | 40.16 |
| W/o Constraint | 40.55 | 52.23 | 31.75 | 33.01 | 65.81 | 26.80 | 32.70 | 15.38 | 26.50 | 11.74 | 79.16 | 13.77 | 35.78 |
| W/o Frozen Matrix $A$ | 47.24 | 54.21 | 36.01 | 43.10 | 67.63 | 29.10 | 35.30 | 16.44 | 26.70 | 11.45 | 80.84 | 18.98 | 38.92 |

- **Obs 1 in § E.4.2: Modifying KORE's design leads to a degradation in overall knowledge retention performance.** In Table 18, the ablated versions W/o Augmentation, W/o Constraint, and W/o Frozen Matrix $A$ exhibit overall performance degradations of 11.59, 15.97, and 12.83 respectively compared to KORE. This significant degradation underscores the high efficacy of KORE's design.

- **Obs 2 in § E.4.2: W/o Constraint yields superior knowledge adaptation performance across a wide spectrum of fine-grained knowledge.** In Table 19, W/o Constraint achieves superior knowledge adaptation performance on fine-grained News and Entity types. These gains stem from KORE-AUGMENTATION's ability to perform profound and structured augmentation.

Table 19: Performance comparison of setting ablation on fine-grained knowledge types with LLaVA-v1.5 (7B).

| Method | News | | | | | | | | | | Entity | | | | | | | | | |
|---|---|---|---|---|---|---|---|---|---|---|---|---|---|---|---|---|---|---|---|---|
| | Avg | | PO | | SP | | BU | | HE | | Avg | | CE | | FI | | AL | | WR | |
| | CEM↑ | F1↑ | CEM↑ | F1↑ | CEM↑ | F1↑ | CEM↑ | F1↑ | CEM↑ | F1↑ | CEM↑ | F1↑ | CEM↑ | F1↑ | CEM↑ | F1↑ | CEM↑ | F1↑ | CEM↑ | F1↑ |
| KORE | 34.74 | 42.96 | 23.83 | 32.31 | 46.19 | 50.38 | 34.69 | 45.74 | 33.20 | 45.23 | 26.17 | 39.39 | 27.79 | 42.61 | 26.93 | 34.05 | 16.52 | 29.54 | 28.81 | 43.05 |
| W/o Augmentation | 14.04 | 20.22 | 8.25 | 14.06 | 15.96 | 20.08 | 14.39 | 23.13 | 14.57 | 25.69 | 7.30 | 16.21 | 8.08 | 20.13 | 8.41 | 14.15 | 3.77 | 6.56 | 13.56 | 22.27 |
| W/o Constraint | 38.45 | 45.16 | 25.57 | 32.56 | 46.43 | 50.66 | 41.33 | 51.22 | 36.84 | 45.78 | 28.97 | 42.12 | 29.67 | 44.18 | 30.45 | 37.75 | 20.58 | 33.59 | 28.81 | 40.02 |
| W/o Frozen Matrix $A$ | 36.49 | 43.42 | 25.11 | 31.70 | 46.43 | 50.44 | 37.82 | 48.20 | 36.44 | 46.44 | 27.01 | 39.85 | 28.05 | 42.88 | 27.95 | 34.57 | 19.13 | 30.95 | 28.81 | 39.86 |

## E.5 MORE RESULTS ON COMPARISON WITH GENERAL AUGMENTATION METHODS

Table 20: Performance comparison of different augmentation methods in knowledge retention with LLaVA-v1.5 (7B).

| Method | COM | | OCR | | M-DIS | | INS | M-IDU | MAT | | HAL | | Avg |
|---|---|---|---|---|---|---|---|---|---|---|---|---|---|
| | MME↑ | MM$^B$↑ | SEED$^{B2P}$↑ | OCR$^{VQA}$↑ | SQA↑ | MMMU$^T$↑ | MIA$^B$↑ | MMDU↑ | Math$^T$↑ | Math$^I$↑ | POPE↑ | Hall$^B$↑ | |
| KORE-AUGMENTATION | 40.55 | 52.23 | **31.75** | **33.01** | 65.81 | **26.80** | 32.70 | 15.38 | **26.50** | 11.74 | **79.16** | **13.77** | **46.47** |
| *Augmentation for Text* | | | | | | | | | | | | | |
| Knowledge-Agnostic | **51.67** | **55.33** | 25.99 | 24.77 | 64.38 | 15.20 | **44.37** | **22.41** | 25.20 | 11.74 | 79.04 | 8.40 | 35.71 |
| Knowledge-Aware | 50.02 | 47.68 | 24.95 | 31.25 | 65.75 | 14.80 | 43.59 | 20.72 | 24.20 | 12.07 | 74.05 | 9.24 | 34.86 |
| *Augmentation for Images* | | | | | | | | | | | | | |
| Knowledge-Agnostic | 50.43 | 52.41 | 11.86 | 14.58 | 64.18 | 9.70 | 43.65 | 21.60 | 22.60 | 11.58 | 73.95 | 8.58 | 32.09 |
| Knowledge-Aware | 51.35 | 51.46 | 27.23 | 21.91 | **66.29** | 14.80 | 40.84 | 18.53 | 21.20 | **17.26** | 69.71 | 7.68 | 34.02 |

- **Obs 1 in § E.5: KORE-AUGMENTATION demonstrates absolute comprehensive performance superiority in knowledge retention evaluations.** In Table 20, KORE-AUGMENTATION surpasses the best general augmentation method by a margin of 10.76 in overall performance, demonstrating its substantially superior capability for knowledge retention.
- **Obs 2 in § E.5: KORE-AUGMENTATION demonstrates superior knowledge adaptation performance across a wide spectrum of fine-grained knowledge types.** In Table 21, KORE-AUGMENTATION achieves the best performance on all News and Entity knowledge types, demonstrating its superiority over general augmentation methods for new knowledge injection.

Table 21: Performance comparison of different augmentation methods on fine-grained knowledge types with LLaVA-v1.5 (7B).

| Method | News | | | | | | | | | | Entity | | | | | | | | | |
|---|---|---|---|---|---|---|---|---|---|---|---|---|---|---|---|---|---|---|---|---|
| | Avg | | PO | | SP | | BU | | HE | | Avg | | CE | | FI | | AL | | WR | |
| | CEM↑ | F1↑ | CEM↑ | F1↑ | CEM↑ | F1↑ | CEM↑ | F1↑ | CEM↑ | F1↑ | CEM↑ | F1↑ | CEM↑ | F1↑ | CEM↑ | F1↑ | CEM↑ | F1↑ | CEM↑ | F1↑ |
| KORE-AUGMENTATION | 38.45 | 45.16 | 25.57 | 32.56 | 46.43 | 50.66 | 41.33 | 51.22 | 36.84 | 45.78 | 28.97 | 42.12 | 29.67 | 44.18 | 30.45 | 37.75 | 20.58 | 33.59 | 28.81 | 40.02 |
| *Augmentation for Text* | | | | | | | | | | | | | | | | | | | | |
| Knowledge-Agnostic | 14.59 | 20.11 | 8.52 | 14.84 | 17.05 | 21.34 | 17.16 | 24.56 | 14.57 | 23.34 | 9.37 | 17.99 | 10.52 | 22.06 | 6.59 | 10.74 | 8.12 | 15.00 | 13.56 | 21.25 |
| Knowledge-Aware | 20.19 | 24.99 | 11.37 | 16.28 | 24.55 | 28.48 | 21.96 | 29.00 | 19.03 | 28.94 | 13.37 | 22.17 | 13.92 | 26.33 | 12.95 | 18.16 | 9.57 | 12.99 | 13.56 | 18.90 |
| *Augmentation for Images* | | | | | | | | | | | | | | | | | | | | |
| Knowledge-Agnostic | 18.38 | 22.42 | 10.72 | 14.88 | 22.97 | 26.92 | 20.11 | 26.60 | 19.84 | 27.28 | 12.26 | 19.87 | 12.35 | 23.27 | 13.07 | 16.59 | 10.43 | 16.13 | 15.25 | 14.83 |
| Knowledge-Aware | 17.15 | 23.01 | 9.99 | 14.93 | 19.35 | 24.35 | 18.08 | 25.79 | 16.19 | 27.05 | 11.97 | 20.84 | 12.86 | 24.29 | 13.86 | 19.59 | 7.25 | 11.47 | 15.25 | 21.78 |

## F CONVERGENCE COMPARISON OF VARIOUS METHODS VIA LOSS CURVES.

Figure 10 presents the training loss curves of the six methods, providing an intuitive comparison of their convergence behaviors. Although KORE and the baseline methods use different training datasets, the loss curves reveal that O-LoRA and SEFE fail to fit the EVOKE's knowledge injection dataset. While LoRA, EWC, and Full-FT converge to very low loss values and successfully fit the evoke dataset, their performance in Table 1 indicates poor generalization to new knowledge, suggesting overfitting. In contrast, KORE not only converges effectively on the KORE-74K dataset but also demonstrates strong generalization capabilities for novel knowledge.

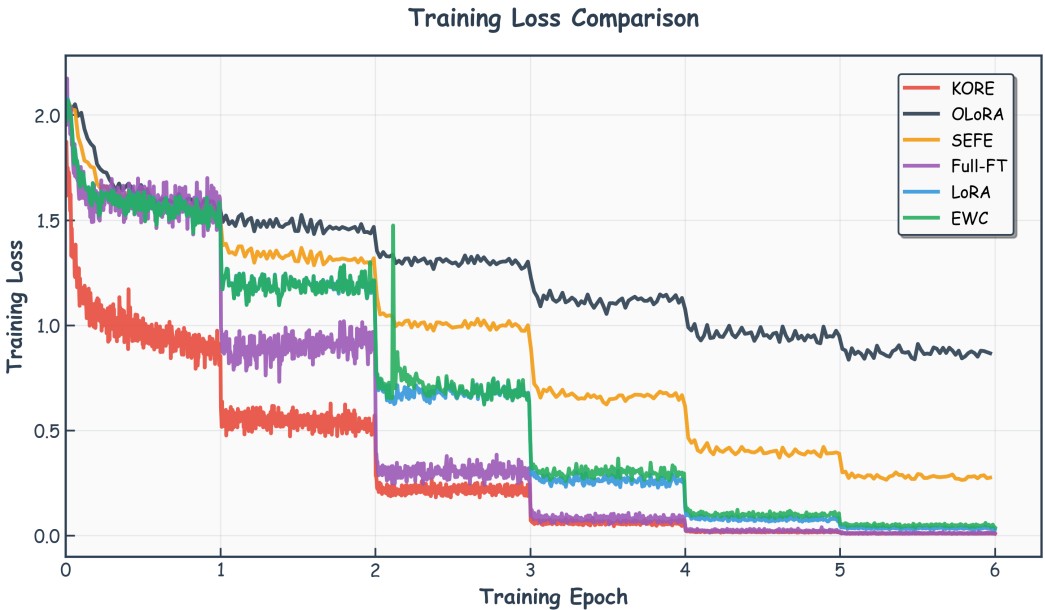

Figure 10: **The training loss curves on EVOKE of Full-FT, LoRA, EWC, O-LoRA, SEFE and KORE.** It should be clarified that Full-FT, LoRA, EWC, O-LoRA, and SEFE are trained using the knowledge injection dataset from EVOKE, whereas KORE is trained using the KORE-74K dataset. The scale of the training data differs between these setups, resulting in varying numbers of iteration steps per epoch. Consequently, KORE exhibits a rapid decrease in loss during the first epoch. The purpose of reporting this loss graph is to provide readers with an intuitive understanding of the convergence of various methods.

## G CASE STUDY

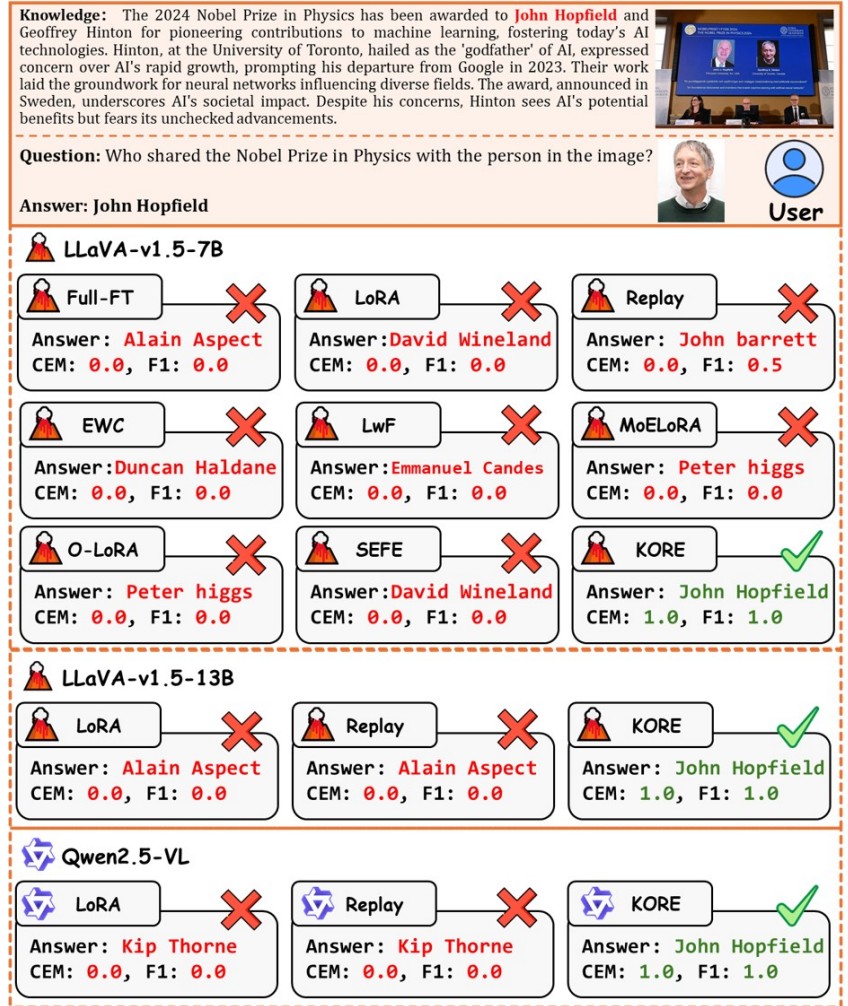

Figure 11: Case Study of News.

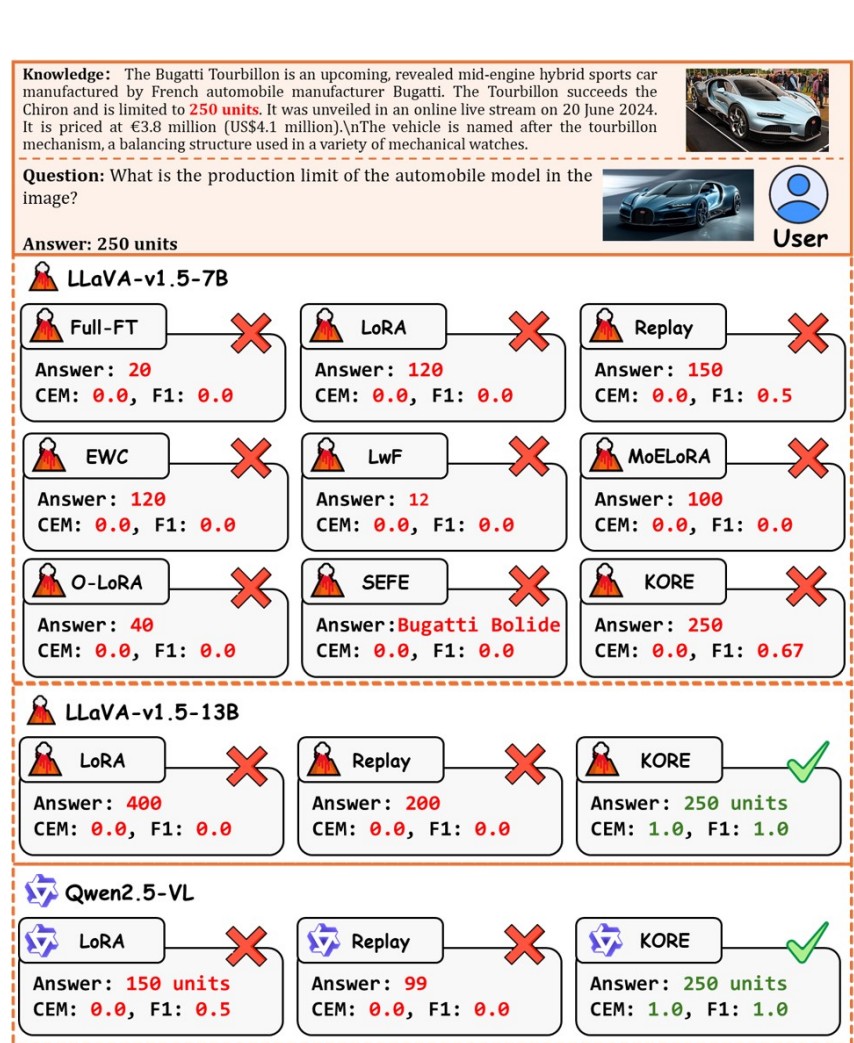

Figure 12: Case Study of Entity.

## H  MORE DETAILS ABOUT KORE-AUGMENTATION

### H.1  MORE CONSTRUCTION PROCESS ABOUT KORE-AUGMENTATION

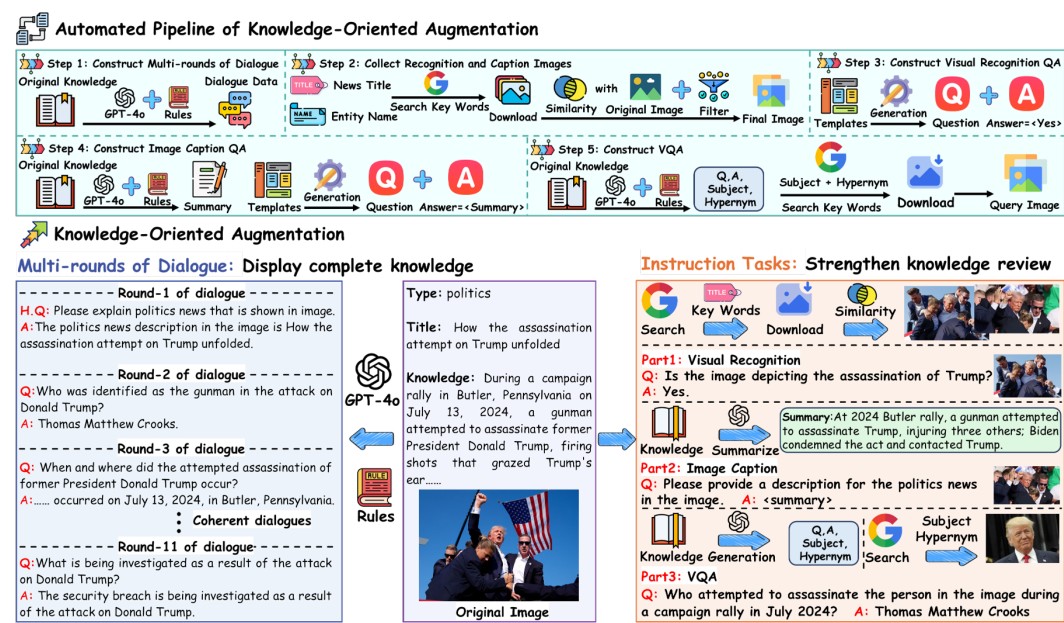

Figure 13: **Overview of construction pipeline for KORE-74K.** The entire data construction process is automated, with only the question templates being manually crafted.

In this section, we elaborate on the implementation of KORE-AUGMENTATION. The fully automated construction pipeline and a data example are illustrated in Figure 13. The following details each step of the pipeline:

- **Step 1: Constructing Multi-rounds of Dialogue.** We design strict rules and diverse task examples, employing GPT-4o to generate multi-turn dialogue data based on the original knowledge. The first turn is a heuristic QA pair randomly selected from templates, such as:
  <''Please explain the {type} news that is shown in the image.'', ''The image provides the following {type} news summary: {title}.''>
  <''Please tell me what the {type} entity in this image is.'', ''The {type} entity shown in the picture is {entity_name}.''>
  The remaining dialogue data are generated automatically by GPT-4o. For each instance, we first generate up to 10 dialogue questions based on the original knowledge and predefined rules. Then, the corresponding answers are produced using the original knowledge, the generated questions, and the rules as input. The query images are taken directly from the original image set. This process results in a complete multi-rounds dialogue dataset, obtaining 9,422 rounds of multi-rounds dialogue data and 75,710 rounds of dialogue. Further templates and prompt designs are provided in § H.3.
- **Step 2: Collecting Recognition and Caption Images.** We use news titles or entity names as search keywords to retrieve and download the top five images via the Google search engine. CLIP (Radford et al., 2021) is then employed to extract visual features from both the downloaded and original images. We compute cosine similarity between them and retain the two images with the highest similarity scores, excluding any identical matches (similarity $\neq$ 1). These selected images serve as query images for visual recognition and image captioning tasks.
- **Step 3: Constructing Visual Recognition QA.** For this task, templates are first manually created. Questions are randomly selected from these templates, and the answer is defined as "Yes". The instruction content is "Answer this question with Yes or No.", and the query image is randomly chosen from the images obtained in Step 2. A template example is provided below:
  <''Is the image depicting news {title}?''>
  <''Can you see {entity_name} in this picture?''>

Further templates and prompt designs are provided in § H.4.

- **Step 4: Constructing Image Caption QA.** We first establish rigorous rules and diverse task examples. Using GPT-4o, we generate summary data based on original knowledge to serve as answers for the image caption task. The instruction content is "Answer this question in one paragraph.', and the query image corresponds to the remaining images from Step 2. Questions are randomly selected from a template, such as:
```
<''Could you please describe the {type} news shown in the picture?''>
<''Please provide a description for the {type} entity in the image.''>
```
Further templates and prompt designs are provided in § H.5.
- **Step 5: Constructing VQA.** First, strict rules and diverse task examples are established. Using GPT-4o, quadruplets ⟨Question, Answer, Subject, Hypernym⟩ are generated based on original knowledge, for instance, <"Who attempted to assassinate the person in the image during a campaign rally in July 2024?", "Thomas Matthew Crooks", "Donald John Trump", "Person">. Subsequently, the subject and hypernym are combined as search keywords to retrieve and download the top 1-ranked image from Google, thereby constructing VQA data. Further prompt designs are provided in § H.6.

Through the above automated pipeline, we have augmented the EVOKE's knowledge injection dataset to KORE-74K, which can better achieve knowledge adaptation.

### H.2 MORE STATISTICAL ANALYSIS ABOUT KORE-AUGMENTATION

In Table 22, we provide detailed statistical data analysis of KORE-74K.

Table 22: **Key Statistics of KORE-74K.**

| Statistic | Number |
|---|---|
| Total data | 74,734 |
|   - Multi-rounds of dialogue data | 9,422 (12.6%) |
|   - Visual recognition data | 9,422 (12.6%) |
|   - Image caption data | 9,422 (12.6%) |
|   - VQA data | 46,468 (62.2%) |
| Number of dialogue rounds | 75,710 |
| Number of unique images | 65,312 |
| Maximum question length | 44 |
| Maximum answer length | 143 |
| Average question length | 15.5 |
| Average answer length | 11.9 |

## H.3 Prompt details regarding Multi-rounds of Dialogue

---

### Prompts and Templates 1 (Part 1): Multi-rounds of Dialogue

***Generation Question Prompt:***

***System Prompt:***

"You have received a descriptive text that provides you with the knowledge, events, and definitions described in the text. You need to generate questions coherently and cover as much of the descriptive text as possible. You just need to output the problem. The maximum number of generated questions is 10. If the previously generated questions are sufficient to cover the entire descriptive text, the output questions can be less than 10."

"From the provided descriptive text, create up to 10 coherent questions that comprehensively cover its content. Your output should consist only of the questions. It is acceptable to generate fewer than 10 questions if the material has been fully covered."

"You are required to formulate a set of coherent questions from a given descriptive text, covering its contents as completely as possible. The number of questions must not exceed 10, but it is permissible to output fewer if they adequately cover the text. The sole output should be the questions."

"Generate a series of logical questions that cover all the knowledge, events, and definitions in the descriptive text you have received. While the maximum number of questions is 10, you can output a smaller number if the text is fully addressed. Please ensure you only output the questions."

"Your task is to generate questions based on a descriptive text, ensuring they are coherent and cover its knowledge, events, and definitions as thoroughly as possible. You should generate a maximum of 10 questions and only output the questions themselves. You may provide fewer than 10 if they are sufficient to cover the entire text."

***User Prompt:***

"News: {news} Please generate questions."
"Given the news: {news} Please generate questions."
"Can you generate questions for the following news: {news}."
"Generate questions for the following news: {news}."
"Please generate questions based on the following news: {news}."

---

## Prompts and Templates 1 (Part 2): Multi-rounds of Dialogue

**Generation Answer Prompt:**

**System Prompt:**

"You have gained knowledge and a problem to be solved. You need to answer this question based on the content of your knowledge. Output your answer."

"You now have the necessary knowledge and a specific problem. Based only on this information, provide your answer to the question and output the result."

"You are equipped with the required information and a problem to resolve. Formulate your answer based solely on the content of this knowledge and then output it."

"Using the knowledge you have been given, solve the problem presented. Your response must be based exclusively on this information. Please output your answer."

"Now that you have the relevant knowledge and the question, you must provide a solution. Ensure your answer is derived strictly from the provided content, then output your response."

**User Prompt:**

"Given the knowledge: {knowledge} Answer the following question: {question}."

"Knowledge: {knowledge} Answer the following question: {question}."

"Answer the following question based on the knowledge: Knowledge:{knowledge} Question: {question}."

"Here is some knowledge: {knowledge} nNow, answer the following question: {question}."

"You are given the knowledge:{knowledge} Can you answer the following question:{question}."

## Prompts and Templates 1 (Part 3): Multi-rounds of Dialogue

**Heuristic question templates for News:**
"What is the {type} news in the image about?"
"Could you summarize the {type} news story presented in the image?"
"What is the {type} news event being depicted in this picture about?"
"Please explain the {type} news that is shown in the image."
"Can you tell me what the {type} news in this image is about?"

**Heuristic answer templates for News:**
"The {type} news description in the image is {title}."
"The {type} news in the image can be described as {title}."
"According to the image, the {type} news description is {title}."
"The image provides the following {type} news summary: {title}."
"The {type} news content shown in the picture is {title}."

**Heuristic answer templates for Entity:**
"What is the {type} entity in the image?"
"Can you identify the {type} entity shown in the picture?"
"What is the {type} entity depicted in this image?"
"Please tell me what the {type} entity in this image is."
"What {type} entity is visible in the photo?"

**Heuristic answer templates for Entity:**
"The {type} entity in the image is {entity_name}."
"The {type} entity shown in the picture is {entity_name}."
"The {type} entity depicted in the image is {entity_name}."
"The {type} entity illustrated in the picture is {entity_name}."
"The {type} entity present in the image is {entity_name}."

## H.4 Prompt details regarding Visual Recognition QA

---

**Prompts and Templates 2: Visual Recognition QA**

***Question templates for News:***
"Is the image depicting news {title}? Answer this question with Yes or No."
"Does this image illustrate the news titled {title}? Answer this question with Yes or No."
"Is this picture related to the news with the headline {title}? Answer this question with Yes or No."
"Is the image about the news report named {title}? Answer this question with Yes or No."
"Does this photo correspond to the news {title}? Answer this question with Yes or No."

***Question templates for Entity:***
"Is {entity_name} in the image? Answer this question with Yes or No."
"Does the image show {entity_name}? Answer this question with Yes or No."
"Can you see {entity_name} in this picture? Answer this question with Yes or No."
"Is {entity_name} visible in the image? Answer this question with Yes or No."
"Does this picture contain {entity_name}? Answer this question with Yes or No."

---

## H.5   PROMPT DETAILS REGARDING IMAGE CAPTION QA

---

**Prompts and Templates 3 (Part 1): Image Caption QA**

***Question templates for News:***
"Please provide a description for the {type} news in the image. Answer this question in one paragraph."
"Could you please describe the {type} news shown in the picture? Answer this question in one paragraph."
"Please offer a description of the {type} news depicted in the image. Answer this question in one paragraph."
"Please give a description of the {type} news depicted here. Answer this question in one paragraph."
"Can you tell me about the {type} news featured in the photograph? Answer this question in one paragraph."

***Answer templates for News:***
"The image depicts {title}. {summary}"

***Question templates for Entity:***
"Please provide a description for the {type} entity in the image. Answer this question in one paragraph."
"Could you please describe the {type} entity shown in the picture? Answer this question in one paragraph."
"Please offer a description of the {type} entity depicted in the image. Answer this question in one paragraph."
"Please give a description of the {type} entity depicted here. Answer this question in one paragraph."
"Can you tell me about the {type} entity featured in the photograph? Answer this question in one paragraph."

***Answer templates for Entity:***
"The image depicts {entity_name}. {summary}"

---

**Prompts and Templates 3 (Part 2): Image Caption QA**

***Generation Summary Prompt:***

***System Prompt:***

"You have acquired a piece of knowledge, and now you need to condense it into a paragraph of no more than 25 words, while trying to maintain the original meaning of the knowledge as much as possible."

"Your task is to take a piece of knowledge you've learned and summarize it. The summary must be a paragraph of 25 words or less, while retaining the original meaning."

"You need to distill the information you have acquired into a concise paragraph. Ensure it does not exceed 25 words and preserves the essence of the original knowledge as accurately as possible."

"Condense a concept you have just learned into a brief paragraph. You must adhere to a 25-word limit, all while making sure the core message remains intact."

"Take the new information you possess and shorten it into a single paragraph. This condensed version must be under 25 words and should accurately reflect the original meaning."

***User Prompt:***

"Knowledge: {knowledge} Please summarize this knowledge."

"Given the knowledge: {knowledge} Please summarize this knowledge."

"Can you summarize this content for the following knowledge: {knowledge}."

"Summarize questions for the following knowledge: {knowledge}."

"Please summarize this content based on the following knowledge: {knowledge}."

## H.6 PROMPT DETAILS REGARDING VQA

---

### Prompts and Templates 4: VQA

***Generation Quadruplets Prompt:***

***System Prompt:***

"You have acquired a piece of knowledge and are now required to generate up to 5 questions based on it. For each generated item, you must provide the question itself, its answer (which should be a word or short phrase), a subject object extracted from the question, and that subject's hypernym. When extracting the subject object, you must follow a critical rule: the subject must be a specific entity that is explicitly mentioned within the question itself, serving as a key reference point. Crucially, this extracted subject cannot be the answer to the question. A helpful test for identifying the correct subject is to check if its name could be logically replaced by a placeholder, such as this company or the entity in the image, while the question remains coherent. If the provided knowledge is fully covered by fewer than 5 questions, you may generate fewer."

"Your task is to generate up to five question sets from the provided knowledge. Each set must include the question, a brief answer (word/phrase), a subject object, and its hypernym. When selecting the subject object, you must follow a key rule: it must be a specific entity explicitly named in the question and cannot be the answer. A good test is to see if a placeholder like this entity can logically replace it. Fewer than five questions are fine if the knowledge is fully covered."

"Based on the knowledge you've acquired, create a maximum of five questions. For each, provide a short answer, identify a subject object, and state its hypernym. The 'subject object' must adhere to this critical constraint: it must be a specific entity mentioned directly in the question that serves as a reference point but is not the answer. To verify your choice, check if substituting a generic term like this item would keep the question coherent. You may generate fewer questions if they are sufficient."

"You are required to produce up to five questions from the given information. For each item, output the question, its short answer, a subject object, and that subject's hypernym. The rule for extracting the subject object is that it must be a specific, named entity within the question's text and must be different from the answer itself. A helpful check is to replace its name with a placeholder (*e.g.,* this organization) to see if the question still makes sense. Fewer questions are acceptable if the topic is fully addressed."

"Formulate as many as five questions based on the knowledge. Each output must consist of the question, a concise answer, an extracted subject object, and its hypernym. A crucial guideline applies: the subject object must be a specific entity named in the question that the query revolves around, but it cannot be the answer. You can confirm the correct subject by checking if a placeholder such as the specified object could logically take its place. Generating all five questions is not necessary if the knowledge is completely covered."

***User Prompt:***

"Knowledge: {knowledge} Please generate questions, answers, subjects, hypernyms."

"Given the knowledge: {knowledge} Please generate questions, answers, subjects, hypernyms."

"Can you generate questions, answers, subjects, hypernyms for the following knowledge: {knowledge}."

"Generate questions, answers, subjects, hypernyms for the following knowledge: {knowledge}."

"Please generate questions, answers, subjects, hypernyms based on the following knowledge: {knowledge}."

---

# I    THE PROCESS OF SAMPLING USING THE ONEVISION DATASET

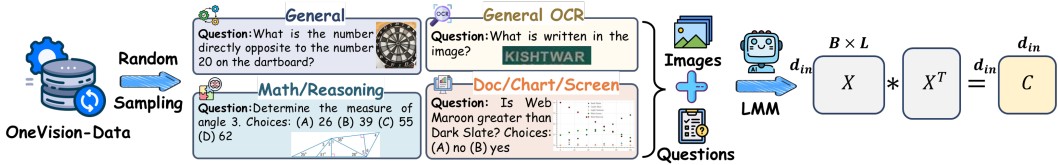

Figure 14: The process of sampling using the OneVision dataset.

# J    HUMAN STUDY

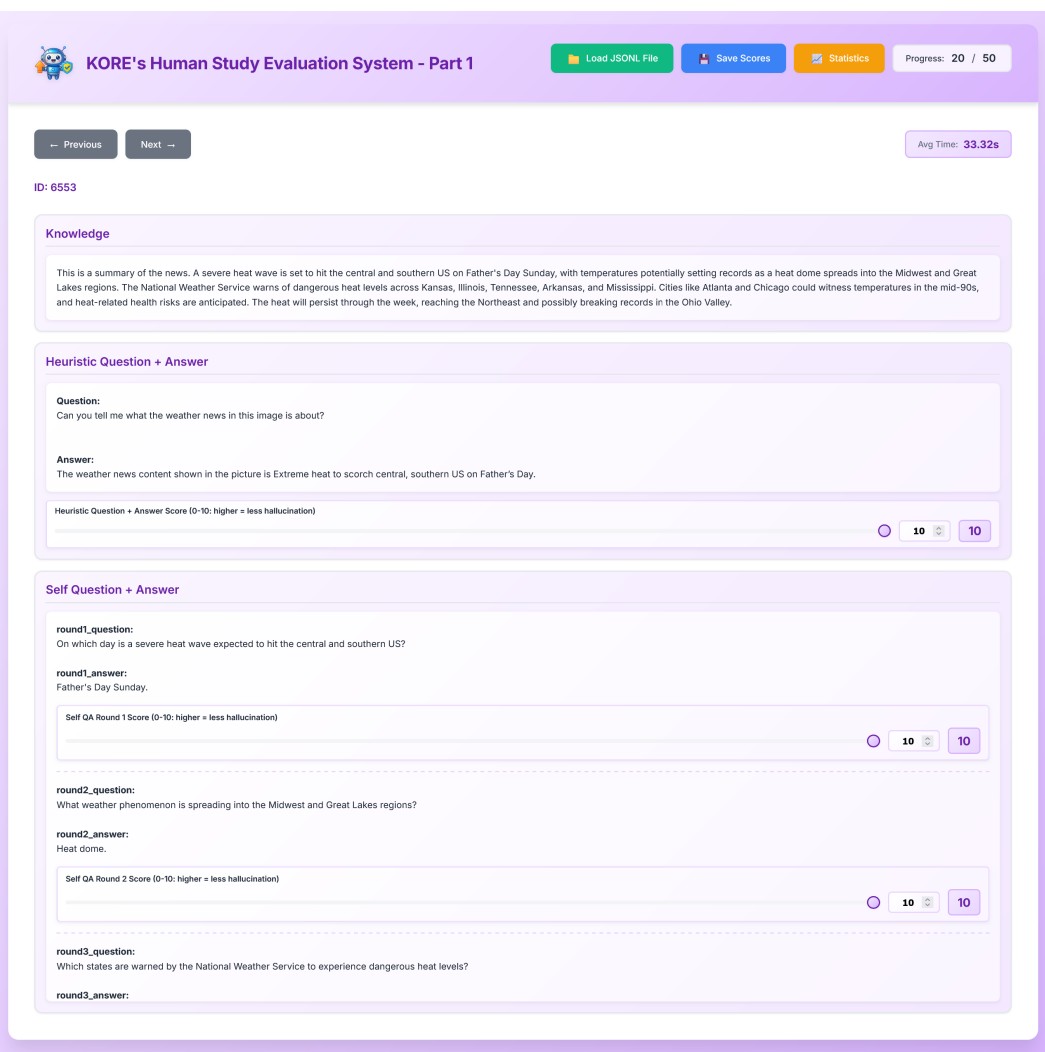

Figure 15: Human study of multi-roundsof dialogue data.

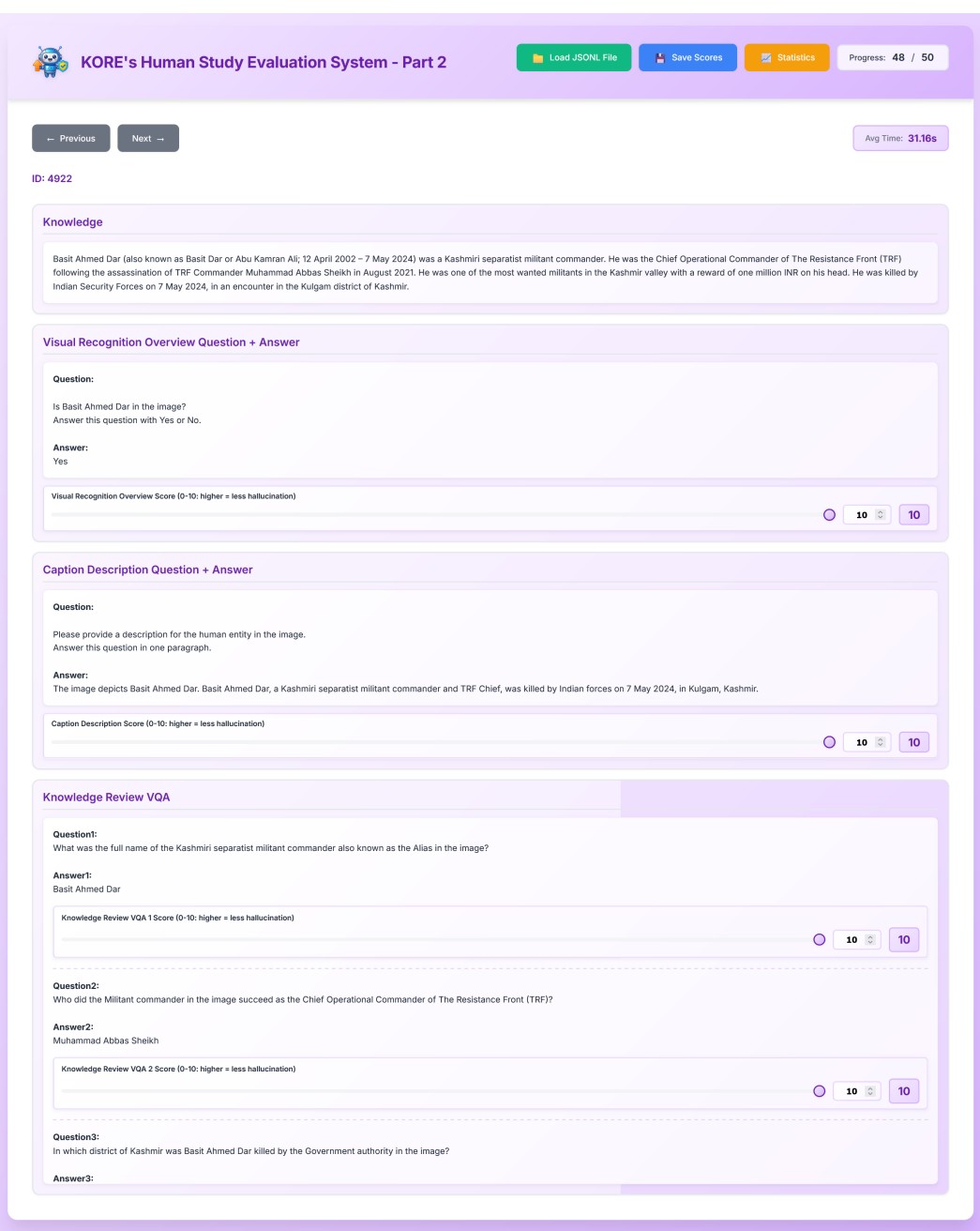

Figure 16: Human study of instruction data.

