# OpenReview forum: "KORE: Enhancing Knowledge Injection for Large Multimodal Models via Knowledge-Oriented Augmentations and Constraints"
_ICLR.cc/2026/Conference — Submitted to ICLR 2026_

### Official Review · Reviewer_Uqia · 2025-10-24

**Soundness:** 2
**Presentation:** 2
**Contribution:** 2
**Rating:** 2
**Confidence:** 4

**Summary:**

The paper proposes a multimodal knowledge injection framework that integrates structured data augmentation with a null-space constraint. It aims to balance new knowledge learning and old knowledge preservation through automatic multimodal task generation and covariance-based regularization.

**Strengths:**

The paper is visually well-presented with clear figures and tables, and the writing is coherent and easy to follow.

**Weaknesses:**

1.	The method uses GPT-4o to generate augmented data, which introduces the risk of incorporating external knowledge. This may lead to unfair comparisons with baselines that do not rely on external models. In other words, the performance gains could partly result from distillation from GPT-4o rather than from the proposed method itself.
2.	The “Knowledge-Oriented Constraint” closely resembles AlphaEdit’s [1] null-space approach. The paper should clarify its conceptual or technical novelty beyond AlphaEdit.
3.	KORE-Augmentation and KORE-Constraint operate independently without clear interaction, making the framework appear as two parallel components rather than an integrated system.
4.	Several conceptually or methodologically related works were not discussed. The paper should include an analysis and discussion with [1][2][3].
5.	The presentation of the theorems in the paper is not standardized and is difficult to follow. For example, in Theorem 1, the symbols are not defined within the statement itself but are introduced later in the proof. The theorem statements should be formalized and rewritten in a clearer, more rigorous manner.

[1] AlphaEdit: Null-Space Constrained Knowledge Editing for Language Models, ICLR

[2] LoRASculpt: Sculpting LoRA for Harmonizing General and Specialized Knowledge in Multimodal Large Language Models, CVPR

[3] LoRI: Reducing Cross-Task Interference in Multi-Task Low-Rank Adaptation, COLM

**Questions:**

1. The roles of the two theorems in the paper are unclear; their purpose and contribution to the overall method are not explicitly explained.
2. The LoRA rank used in this paper appears relatively high compared to prior works. How does the proposed method perform under lower-rank settings?
3. In Table 5, the W/o Constraint variant performs about 3% better than KORE on the EVOKE dataset. Does this indicate that the proposed constraint may negatively affect performance on EVOKE?

---

> ### Author Response · Authors · 2025-11-25
> **Response to reviewer Uqia (1/6)**
>
> Dear Reviewer Uqia:
>
> Thank you for your positive feedback and valuable suggestions. We sincerely appreciate the time and effort you have dedicated to reviewing our work. Below, we meticulously provide responses to each of your comments and outline the modifications based on your suggestions. All revisions are highlighted in blue.
>
> ---
> > **W1: The method uses GPT-4o to generate augmented data, which introduces the risk of incorporating external knowledge. This may lead to unfair comparisons with baselines that do not rely on external models. In other words, the performance gains could partly result from distillation from GPT-4o rather than from the proposed method itself.**
>
>
> We will clarify from the following aspects:
>
>
> - Our method primarily uses GPT-4o for the structured augmentation of raw knowledge within the EVOKE dataset, helping the model better internalize knowledge rather than modifying knowledge capacity. Therefore, **GPT-4o merely acts as the executor of KORE-Augmentation and is not the provider of the knowledge itself**. (Section 3.1, Page 3-4 and Appendix H.1, Page 30-31)
>     - The raw knowledge in the EVOKE dataset is **inherently sufficient to answer the test questions**, making external knowledge unnecessary.
>     - If external knowledge is indeed introduced during this process, it could introduce erroneous or conflicting information, consequently leading to a decline in knowledge adaptation performance.
> - Prior to augmenting the EVOKE dataset with GPT-4o, **we recognize the importance of ensuring the semantic fidelity of the augmented knowledge and avoiding bias.** Therefore, we write 5 strict quality prompts for each augmentation module to guarantee task clarity and prompt diversity. We now present partial examples of the prompts for each augmentation module.
>     - **Multi-rounds of Dialogue:** "**From the provided descriptive text**, create up to 10 coherent questions that comprehensively cover its content." `Appendix H.3, Page 32`  and "You need to answer this question **based on the content of your knowledge**." `Appendix H.3, Page 33`
>     - **Image Caption QA:** "The summary **must be a paragraph of 25 words or less, while retaining the original meaning**." `Appendix H.5, Page 35`
>     - **VQA:** "**Based on the knowledge you’ve acquired**, create a maximum of five questions." `Appendix H.6, Page 36`
> - **Human studies:** We randomly select 50 data samples and ask two annotators to score their quality on a scale of 0 to 10, where a higher score indicates higher quality. The scoring criteria are: **1)** the augmented data content must be derived entirely from the original knowledge, with no introduction of external knowledge; and **2)** the augmented data content must be semantically consistent with the original knowledge. The human study results below demonstrate that the augmented data quality is extremely high and introduces no external knowledge. The front-end webpage examples used for the human study is in Appendix J. `Page 37-38, Figure 15-16`
>
>
>
>
> |  |  | Multi-rounds of Dialogue Data |  | Instruction Tasks Data |  |  |  |
> | :--- | :---: | :---: | :---: | :---: | :---: | :---: | :---: |
> |  |  | Heuristic Q&A | Multi-rounds of Dialogue Q&A | Visual Recognition | Image Caption | VQA |  |
> | Score↑ |  | 9.76 ± 0.01 | 9.78 ± 0.03 | 9.91 ± 0.02 | 9.80 ± 0.05 | 9.81 ± 0.04 |  |
>
>
>
> - **Experimental verification:** In Section 4.5 (Page 9, Line 464-476), we present a performance comparison between KORE and general augmentation methods, where Textual augmentation is also performed using GPT-4o.
>     - From the table below, we observe that the performance of Textual Knowledge-Aware augmentation, Textual Knowledge-Agnostic augmentation, and KORE-Augmentation all surpass that of LoRA, indicating that the use of GPT-4o does not introduce conflicting external knowledge.
>     - KORE-Augmentation's performance far surpasses that of Textual Knowledge-Aware augmentation and Textual Knowledge-Agnostic augmentation. This constitutes a direct and fair comparison, proving that our method's performance gain does not stem from distillation of GPT-4o.
>
> | Method | K.A | CEM | F1 |
> | :--- | :---: | :---: | :---: |
> | LoRA | 14.67 | 15.23 | 18.31 |
> | Textual Knowledge-Aware augmentation | **20.29** | 16.94 | 23.64 |
> | Textual Knowledge-Agnostic augmentation | **15.60** | 12.10 | 19.10 |
> | KORE-Augmentation | **38.82** | 33.93 | 43.71 |

---

> ### Author Response · Authors · 2025-11-25
> **Response to reviewer Uqia (2/6)**
>
> > **W2: The “Knowledge-Oriented Constraint” closely resembles AlphaEdit’s null-space approach. The paper should clarify its conceptual or technical novelty beyond AlphaEdit.**
>
> We appreciate the opportunity to clarify our position and contributions. Our contributions are distinct in terms of objectives, architecture, and task applicability:
>
> - **Null space is a widely recognized standard tool:** Although both KORE and AlphaEdit employ the null space to mitigate interference with prior knowledge, this does not diminish KORE's originality and novelty. This is because the null space is a widely recognized standard tool used in diverse fields such as security and alignment[1], mitigating object hallucination[2], multi-agent collaborative control[3], and outlier detection[4].
>
> [1] A Guardrail for Safety Preservation: When Safety-Sensitive Subspace Meets Harmful-Resistant Null-Space.
> [2] Nullu: Mitigating Object Hallucinations in Large Vision-Language Models via HalluSpace Projection.
> [3] Null space-based behavioral control applied to a formation of two quadrotors transporting a cable suspended load.
> [4] Outlier Detection through Null Space Analysis of Neural Networks.
>
> - **The difference in using null space between KORE and AlphaEdit:**
>     - **Multimodal null space:** AlphaEdit uses 100,000 text triplets from Wikipedia to extract its null space, only focusing on the purely textual domain. In stark contrast, KORE utilizes a minimal set of only 256 QA pairs from the OneVision dataset to extract the multimodal null space required to protect prior knowledge and capabilities. KORE achieves excellent knowledge retention performance with a significantly smaller data volume (details in Section 4.1, Table 1). Furthermore, KORE adopts the null-space concept only after rigorous experimental validation in the multimodal setting, ensuring it is not merely a blind imitation (details in Section 3.3, Figure 4, and Appendix D, line xx).
>     - **Simpler constraints:** AlphaEdit requires null-space constraints during the optimization of every edited knowledge instance **($W + \triangle P$)**, whereas KORE only requires a null-space constraint during the initialization of the LoRA's low-rank matrices $A$ and $B$ **($W_0 + BA$)**. This merely defines a fine-tuning direction that minimizes interference with prior knowledge throughout the subsequent optimization. Consequently, KORE's operation is simpler and more convenient.
>     - **Better scalability and flexibility:** AlphaEdit uses text triplet data from Wikipedia, while KORE uses QA pairs from the OneVision dataset, which imposes fewer data format constraints. Furthermore, KORE can easily implement specific knowledge-oriented constraints based on user needs (Section 4.2, Page 7-8, Line 368-374+392-402), whether for specific capabilities or factual knowledge, demonstrating extremely strong scalability and flexibility.
>     - **Synergistic optimization objectives:** AlphaEdit primarily focuses on using the null space to constrain the interference with prior knowledge during new knowledge updates, with less attention paid to the new knowledge update performance. However, both knowledge editing and knowledge injection require synergistically optimizing the goals of learning new knowledge(knowledge adaptation) and preserving old knowledge(knowledge retention). KORE, using the knowledge-oriented approach as its pivot, synergistically optimizes the balance between knowledge adaptation and retention at different stages of knowledge injection, which better aligns with practical application requirements.
>
> - **Summary of our contribution:** KORE is the first synergistic method of knowledge-oriented augmentations and constraints for injecting new knowledge into large multimodal models while preserving old knowledge. KORE-Augmentation (for accurate adaptation) and KORE-Constraint (for powerful retention) use the knowledge-oriented approach as a pivot to optimize the balance between knowledge adaptation and retention at different stages of knowledge injection. This synergistic design is necessary to address the challenges of poor generalization and catastrophic forgetting (in teaser, Page 1, Figure 1) in multimodal knowledge injection scenarios, which prior continual learning and methods like AlphaEdit cannot solve alone.

---

> ### Author Response · Authors · 2025-11-25
> **Response to reviewer Uqia (3/6)**
>
> > **W3 (Part 1/2): KORE-Augmentation and KORE-Constraint operate independently without clear interaction, making the framework appear as two parallel components rather than an integrated system.**
>
>
> We will clarify from the following aspects:
>
> - KORE-Augmentation and KORE-Constraint do not operate entirely independently; rather, they interact **at different stages of the knowledge injection process**, making both components indispensable.
> - In the knowledge injection scenario, the model must simultaneously inject new knowledge and preserve prior knowledge and capabilities, which represent distinct levels of objectives. We therefore employ **knowledge-oriented control strategy as the pivot**, utilizing KORE-Augmentation and KORE-Constraint at different stages of the injection process to achieve these goals. During the initialization phase, KORE-Constraint defines a fine-tuning direction for the model that minimizes interference with prior knowledge and capabilities. Subsequently, in the knowledge injection phase, KORE-Augmentation transforms the knowledge to be learned into profound and structured knowledge, helping the model better internalize the information. Crucially, KORE focuses on optimizing the balance between knowledge adaptation and retention, rather than maximizing any single objective.
> - **Ablation experiments for verification:**
>     - We present the performance comparison of KORE against the W/o Constraint and W/o Augmentation variants in the table below. Based on Tables 1 and 2, we find that the W/o Augmentation variant shows a significant drop in knowledge adaptation performance across eight fine-grained knowledge types , and the W/o Constraint variant exhibits performance degradation across 11 knowledge retention benchmarks. This is sufficient evidence that KORE operates as an indispensable whole, where neither component can be omitted.
>
> > **Table 1: Performance comparison on fine-grained knowledge types.**
>
> | Method |  | News |  |  |  |  |  |  |  | Entity   | | |  |  |  |  |
> |:---|:---:|:---:|:---:|:---:|:---:|:---:|:---:|:---:|:---:|:---:|:---:|:---:|:---:|:---:|:---:|:---:|
> |  |  | Politics |  | Sports |  | Business |  | Health |  | Celebrity |  | Film |  | Album |  | WrittenWork |  |
> |  |  | CEM↑ | F1↑ | CEM↑ | F1↑ | CEM↑ | F1↑ | CEM↑ | F1↑ | CEM↑ | F1↑ | CEM↑ | F1↑ | CEM↑ | F1↑ | CEM↑ | F1↑ |
> | KORE |  | 23.83 | 32.31 | 46.19 | 50.38 | 34.69 | 45.74 | 33.20 | 45.23 | 27.79 | 42.61 | 26.93 | 34.05 | 16.52 | 29.54 | 28.81 | 43.05 |
> | W/o Augmentation |  | 8.25 | 14.06 | 15.96 | 20.08 | 14.39 | 23.13 | 14.57 | 25.69 | 8.08 | 20.13 | 8.41 | 14.15 | 3.77 | 6.56 | 13.56 | 22.27 |
> | Decline(%) |  | **65.38↓** | **56.48↓** | **65.45↓** | **60.14↓** | **58.52↓** | **49.43↓** | **56.11↓** | **43.20↓** | **70.92↓** | **52.76↓** | **68.77** | **58.44↓** | **77.18↓** | **77.79↓** | **52.93↓** | **48.27↓** |
>
> > **Table 2: Performance comparison on fine-grained knowledge retention evaluations.**
>
> | Method | MME | MMBench | SEEDBench2Plus | OCRVQA | ScienceQA | MMMU | MIA-Bench | MMDU | MathVista | MathVision | POPE | HallusionBench | Avg |
> | :--- | :---: | :---: | :---: | :---: | :---: | :---: | :---: | :---: | :---: | :---: | :---: | :---: | :---: |
> | KORE | 49.84 | 54.98 | 37.73 | 44.24 | 68.06 | 29.30 | 38.54 | 16.58 | 25.10 | 12.09 | 80.99 | 22.51 | 40.00 |
> | W/o Constraint | 40.55 | 52.23 | 31.75 | 33.01 | 65.81 | 26.80 | 32.70 | 15.38 | 26.50 | 11.74 | 79.16 | 13.77 | 35.78 |
> | Decline or Increase(%) | **18.64↓** | **5.00↓** | **15.85↓** | **25.38↓** | **3.31↓** | **8.53↓** | **15.15↓** | **7.24↓** | 5.58↑ | **2.89↓** | **2.26↓** | **38.83↓** | **10.53↓**|

---

> ### Author Response · Authors · 2025-11-25
> **Response to reviewer Uqia (4/6)**
>
> > **W3 (Part 1/2): KORE-Augmentation and KORE-Constraint operate independently without clear interaction, making the framework appear as two parallel components rather than an integrated system.**
>
> - **Combination experiments for verification:**
>     - We also combine MoELORA and Replay with general augmentation methods to test the feasibility of simple hybrid strategies in knowledge injection (Table 3, the second row of value for each combination method represents the percentage of decline or increase compared to MoELORA/Replay performance), thereby demonstrating that KORE is not merely a simple combination-style method.
>     - The knowledge adaptation performance of MoELORA when combined with any augmentation method improves, but **its performance across all seven knowledge retention evaluations decreases, leading to an overall decline in performance**. We believe this occurs because knowledge augmentation overloads the expert modules by forcing them to learn multiple, complex, and semantically diverse knowledge variations. This saturation causes the separation mechanism of the experts to fail, resulting in a significant drop in knowledge retention performance.
>     - The knowledge adaptation performance of Replay **significantly declines after combining it with augmentation methods**. We believe this happens because Replay and augmentation methods are both data-level approaches, and they mutually dilute each other's data distribution, leading to a decline in overall performance.
>     - The performance loss resulting from combining MoELORA and Replay with general augmentation methods demonstrates that simple method combinations often introduce conflicts and interference. Conversely, KORE is not a simple combination of methods; KORE-Augmentation (for accurate adaptation) and KORE-Constraint (for powerful retention) use the knowledge-oriented approach as a pivot to optimize the balance between knowledge adaptation and retention at different stages of knowledge injection.
>
>
> > **Table 3: Performance comparison about combination previous continual learning methods with general augmentation methods.**
>
> | Method | CEM↑ | F1↑ | COM↑ | OCR↑ | M-DIS↑ | INS↑ | M-IDU↑ | MAT↑ | HAL↑ | Avg↑ |
> | :--- | :---: | :---: | :---: | :---: | :---: | :---: | :---: | :---: | :---: | :---: |
> | **MoELoRA** | 7.12 | 12.60 | 62.60 | 39.10 | 48.90 | 64.97 | 18.66 | 19.25 | 51.22 | **26.69** |
> | MoELoRA+Textual Knowledge-Aware | 15.67 | 19.60 | 49.55 | 31.39 | 43.89 | 36.62 | 15.42 | 18.78 | 42.77 | 25.85 |
> | **Decline or Increase(%)** | 120.08↑ | 55.56↑ | **20.84↓** | **19.72↓** | **10.24↓** | **43.64↓** | **17.38↓** | **2.42↓** | **16.49↓** | **3.17↓** |
> | MoELoRA+Textual Knowledge-Agnostic | 8.67 | 13.84 | 46.95 | 24.24 | 43.51 | 43.33 | 16.11 | 18.17 | 37.66 | 22.05 |
> | **Decline or Increase(%)** | 21.77↑ | 9.84↑ | **25.00↓** | **38.00↓** | **11.02↓** | **33.31↓** | **13.67↓** | **5.61↓** | **26.46↓** | **17.38↓** |
> | MoELoRA+image Knowledge-Aware | 14.90 | 19.64 | 52.11 | 29.57 | 43.76 | 30.35 | 12.69 | 16.89 | 43.71 | 25.00 |
> | **Decline or Increase(%)** | 109.27↑ | 55.87↑ | **16.76↓** | **24.37↓** | **10.51↓** | **53.29↓** | **31.98↓** | **12.27↓** | **14.65↓** | **6.35↓** |
> | MoELoRA+image Knowledge-Agnostic | 13.78 | 19.40 | 52.53 | 32.51 | 44.70 | 32.93 | 11.58 | 16.88 | 44.24 | 25.11 |
> | **Decline or Increase(%)** | 93.54↑ | 53.97↑ | **16.09↓** | **16.86↓** | **8.59↓** | **49.32↓** | **37.94↓** | **12.34↓** | **13.63↓** | **5.95↓** |
> | **Replay** | 11.36 | 17.98 | 59.72 | 38.04 | 48.64 | 62.33 | 19.31 | 19.17 | 51.67 | **28.68** |
> | Replay+Textual Knowledge-Aware | 10.42 | 17.42 | 58.93 | 43.45 | 48.27 | 62.15 | 23.47 | 18.20 | 52.25 | 28.87 |
> | **Decline or Increase(%)** | **8.27↓** | **3.11↓** | **1.32↓** | 14.23↑ | **0.75↓** | **0.30↓** | 21.52↑ | **5.04↓** | 1.12↑ | 0.64↑ |
> | Replay+Textual Knowledge-Agnostic | 8.75 | 16.08 | 57.89 | 34.62 | 43.35 | 58.88 | 19.86 | 17.87 | 51.39 | 26.48 |
> | **Decline or Increase(%)** | **22.98↓** | **10.57↓** | **3.06↓** | **8.99↓** | **10.87↓** | **5.54↓** | 2.85↑ | **6.77↓** | **0.53↓** | **7.67↓** |
> | Replay+image Knowledge-Aware | 10.34 | 17.19 | 58.76 | 32.48 | 49.23 | 58.41 | 14.24 | 17.73 | 52.63 | 27.13 |
> | **Decline or Increase(%)** | **8.98↓** | **4.39↓** | **1.61↓** | **14.60↓** | 1.23↑ | **6.29↓** | **26.27↓** | **7.47↓** | 1.87↑ | **5.41↓** |
> | Replay+image Knowledge-Agnostic | 9.61 | 16.41 | 60.16 | 44.95 | 47.97 | 62.63 | 11.36 | 17.65 | 52.57 | 27.74 |
> | **Decline or Increase(%)** | **15.40↓** | **8.73↓** | 0.73↑ | 18.18↑ | **1.37↓** | 0.47↑ | **41.17↓** | **7.89↓** | 1.75↑ | **3.29↓** |

---

> ### Author Response · Authors · 2025-11-25
> **Response to reviewer Uqia (5/6)**
>
> > **W4: Several conceptually or methodologically related works were not discussed. The paper should include an analysis and discussion with.**
>
> We compare and analyze the differences among AlphaEdit, LoRASculpt, LoRI, and KORE from the following aspects:
>
> - AlphaEdit[1] introduces null space constraints for knowledge editing, focusing on precisely modifying specific factual knowledge, but it is limited to the text domain. Null space constraint is a standard tool explored in several prior works, our contribution lies in leveraging knowledge-oriented control to resolve the challenges of knowledge injection and catastrophic forgetting in LMMs, leading to accurate adaptation and powerful retention.Our evaluation of forgetting is significantly more comprehensive, employing 12 benchmarks across seven capability dimensions.
> - LoRASculpt[2] aims to harmonize general and specialized knowledge in LMMs by optimizing and regulating the LoRA adapter to prevent specialized fine-tuning from damaging general capabilities. Our contribution, in contrast, proposes a synergistic method of knowledge-oriented augmentations and constraints for injecting new knowledge into large multimodal models while preserving old knowledge. We demonstrate its effectiveness through extensive experiments. Furthermore, while LoRASculpt is primarily evaluated using VQA data, KORE focuses on the more challenging task of injecting factual knowledge.
> - LoRI[3] primarily addresses cross-task interference in the multi-task learning (MTL) setting, utilizing precise and non-redundant parameter updates to enhance multi-task performance while maintaining core model capabilities, but it does not involve multimodal scenarios. In contrast, KORE is designed for the continual learning setting to manage the interference between new and old knowledge, ensuring that a model's existing knowledge and capabilities experience minimal interference when new factual knowledge is continuously injected, thereby offering new insights for building continually evolving LMMs.
>
> We discuss and reference these works in the `Knowledge Forgetting`(Page 3, Line 116-130) section of related works.
>
> [1] AlphaEdit: Null-Space Constrained Knowledge Editing for Language Models.(ICLR25)
>
> [2] LoRASculpt: Sculpting LoRA for Harmonizing General and Specialized Knowledge in Multimodal Large Language Models.(CVPR25)
>
> [3] LoRI: Reducing Cross-Task Interference in Multi-Task Low-Rank Adaptation.(COLM25)
>
>
>
> ---
> > **W5+Q1: The presentation of the theorems in the paper is not standardized and is difficult to follow. For example, in Theorem 1, the symbols are not defined within the statement itself but are introduced later in the proof. The theorem statements should be formalized and rewritten in a clearer, more rigorous manner.**
>
> We will clarify from the following aspects:
>
>
> - **The role and contribution of two theorems:**
>     - **Theorem 1:** In Section 3.2(Page 4-5, Line 207-252), KORE-Constraint initializes the LoRA's low-rank matrix $A$ within the null space of the covariance matrix $C$, which represents prior knowledge and capabilities. This claim is the premise for KORE-Constraint's validity and effectiveness, and Theorem 1 is used to prove its feasibility.
>     - **Theorem 2:** In KORE, the LoRA's low-rank matrix $A$ is frozen, and only matrix $B$ is fine-tuned during the process. We demonstrate in Theorem 2 why this operation minimizes interference with prior knowledge and capabilities during fine-tuning. Theorem 2 extends Theorem 1: as long as Theorem 1 ensures that matrix $A$ lies in the null space of the covariance matrix $C$, the final output of each layer, $W^* X$, remains approximately equal to $W_0 X$, regardless of how the parameters of matrix $B$ are adjusted.
> - Following your suggestion, we have rewritten the derivation process for both theorems in Appendix C (Page 19-21, Line 1010-1082) in a clearer and more rigorous manner.  We hope this revision meets your approval and aids subsequent reader understanding.

---

> ### Author Response · Authors · 2025-11-25
> **Response to reviewer Uqia (6/6)**
>
> > **Q2: The LoRA rank used in this paper appears relatively high compared to prior works. How does the proposed method perform under lower-rank settings?**
>
> We will clarify from the following aspects:
>
> - Our method uses a larger LoRA rank because the matrix A, initialized by LoRA, is frozen during training, which results in fewer trainable parameters compared to the baselines. Therefore, we adjust the LoRA rank to match the number of trainable parameters with the baselines, ensuring a fair experimental comparison.
> - **We conduct an ablation study on rank in Section 4.3 (Page 9,Figure 7)**, confirming that KORE `rank=64, params=108M` still achieves better comprehensive performance than Replay (the best performing baseline) `rank=128, params=340M`. For convenient reference, we present the relevant results in the table below. (More experimental results are available in Appendix E.4.1, Tables 16–17).
>
>
> | Method | CEM↑ | F1↑ | COM↑ | OCR↑ | M-DIS↑ | INS↑ | M-IDU↑ | MAT↑ | HAL↑ | Avg↑ |
> | :--- | :---: | :---: | :---: | :---: | :---: | :---: | :---: | :---: | :---: | :---: |
> | Replay (rank=128,params=304M) | 11.36 | 17.98 | 59.72 | 37.98 | 48.64 | 62.33 | 19.31 | 19.17 | 51.67 | 28.68 |
> | KORE (rank=64,params=108M) | 24.00 | 33.07 | 45.35 | 29.46 | 45.02 | 44.07 | 19.62 | 18.08 | 44.48 | **31.81** |
> | KORE (rank=128,params=195M) | 30.72 | 40.55 | 49.97 | 36.05 | 47.07 | 34.87 | 10.00 | 17.46 | 50.30 | **35.37** |
> | KORE (rank=235,params=304M) | 30.65 | 41.26 | 52.41 | 40.98 | 48.68 | 38.54 | 16.58 | 18.59 | 51.75 | **37.09** |
> | KORE (rank=256,params=369M) | 31.05 | 41.32 | 52.48 | 39.96 | 48.96 | 60.02 | 23.18 | 18.09 | 51.50 | **39.11** |
>
>
>
>
> ---
> > **Q3: In Table 5, the W/o Constraint variant performs about 3% better than KORE on the EVOKE dataset. Does this indicate that the proposed constraint may negatively affect performance on EVOKE?**
>
> We will clarify from the following aspects:
>
> - In the context of knowledge injection, performance requires a synergistic balance between knowledge adaptation and retention. Therefore, the localized performance gain observed for the W/o Constraint variant in knowledge adaptation does not imply that KORE-Constraint is detrimental to Knowledge Adaptation on EVOKE. **KORE's primary design goal is to optimize the balance between adaptation and retention through the synergy of KORE-augmentation and KORE-Constraint, rather than maximizing any single metric**.
> - Although the W/o Constraint variant brings a slight gain in knowledge adaptation, **this comes at the cost of more severe catastrophic forgetting**, which harms overall objective performance. We present the comparison of performance on knowledge retention in the table below. It is evident that the W/o Constraint variant performs worse than KORE on 11 out of 12 benchmarks, with the exception of MathVision, resulting in an average score decrease of 10.53%.
>
>
> | Method | MME | MMBench | SEEDBench2Plus | OCRVQA | ScienceQA | MMMU | MIA-Bench | MMDU | MathVista | MathVision | POPE | HallusionBench | Avg |
> | :--- | :---: | :---: | :---: | :---: | :---: | :---: | :---: | :---: | :---: | :---: | :---: | :---: | :---: |
> | KORE | 49.84 | 54.98 | 37.73 | 44.24 | 68.06 | 29.30 | 38.54 | 16.58 | 25.10 | 12.09 | 80.99 | 22.51 | 40.00 |
> | W/o Constraint | 40.55 | 52.23 | 31.75 | 33.01 | 65.81 | 26.80 | 32.70 | 15.38 | 26.50 | 11.74 | 79.16 | 13.77 | 35.78 |
> | Decline or Increase(%) | **18.64↓** | **5.00↓** | **15.85↓** | **25.38↓** | **3.31↓** | **8.53↓** | **15.15↓** | **7.24↓** | 5.58↑ | **2.89↓** | **2.26↓** | **38.83↓** | **10.53↓**|
>
>
> **Hope our clarification could resolve your concern.**

---

> ### Author Response · Authors · 2025-11-28
> **Please tell us if any concern remains**
>
> Dear Reviewer Uqia, ﻿
> ﻿
>
>
> We truly appreciate the effort you have invested in reviewing this paper. We have submitted a **detailed rebuttal** addressing each of your points.
>
>
> ﻿**If you find that any explanation is still insufficient or could benefit from further clarification, please tell us and we would be happy to elaborate.**
> ﻿
>
>
> Thanks for your consideration.
>
>
> ﻿Authors of Paper 374

---

### Official Review · Reviewer_WzoY · 2025-10-29

**Soundness:** 3
**Presentation:** 4
**Contribution:** 2
**Rating:** 4
**Confidence:** 4

**Summary:**

This paper proposes the KORE method, which achieves a balanced integration of new and existing knowledge in large models through the collaboration of KORE-AUGMENTATION and KORE-CONSTRAINT. The former automatically generates multi-turn dialogues and multimodal task data to promote knowledge internalization, while the latter uses null-space constraints to prevent forgetting. Experiments demonstrate that KORE significantly enhances the knowledge learning and retention capabilities of multimodal models.

**Strengths:**

- KORE-AUGMENTATION is highly innovative and serves as a reasonable and effective data augmentation approach.
- The method achieves strong empirical results, reaching state-of-the-art performance.

**Weaknesses:**

- The core idea of KORE-CONSTRAINT is quite similar to AlphaEdit [1] (both employ projection onto the null space to mitigate interference with prior knowledge), which weakens the originality of this work.
- Experiments are conducted only on the EVOKE benchmark, while another equally important benchmark in this field, CoIN [2], is neglected.

[1] [2025-ICLR] Alphaedit: Null-space constrained knowledge editing for language models
[2] [2024-NeurIPS] CoIN: A benchmark of continual instruction tuning for multimodel large language models

**Questions:**

- AlphaEdit [1] also employs projection onto the null space to mitigate interference with prior knowledge. Could you clarify how your approach differs from theirs?
- SEFE [2] consists of two components, ASD and RegLoRA. Since the authors of SEFE did not apply ASD to the EVOKE dataset, did you fully reproduce ASD on the EVOKE benchmark for comparison, or are your SEFE results based solely on its RegLoRA component?

[1] [2025-ICLR] Alphaedit: Null-space constrained knowledge editing for language models
[2] [2025-ICML] SEFE: Superficial and Essential Forgetting Eliminator for Multimodal Continual Instruction Tuning

---

> ### Author Response · Authors · 2025-11-25
> **Response to reviewer WzoY (1/2)**
>
> Dear Reviewer WzoY:
>
> Thank you for your positive feedback and valuable suggestions. We sincerely appreciate the time and effort you have dedicated to reviewing our work. Below, we meticulously provide responses to each of your comments and outline the modifications based on your suggestions. All revisions are highlighted in blue.
>
> ---
> > **W1: The core idea of KORE-CONSTRAINT is quite similar to AlphaEdit (both employ projection onto the null space to mitigate interference with prior knowledge), which weakens the originality of this work.**
>
>
>
> We appreciate the opportunity to clarify our position and contributions. Our contributions are distinct in terms of objectives, architecture, and task applicability:
>
> - **Null space is a widely recognized standard tool:** Although both KORE and AlphaEdit employ the null space to mitigate interference with prior knowledge, this does not diminish KORE's originality and novelty. This is because the null space is a widely recognized standard tool used in diverse fields such as security and alignment[1], mitigating object hallucination[2], multi-agent collaborative control[3], and outlier detection[4].
>
> [1] A Guardrail for Safety Preservation: When Safety-Sensitive Subspace Meets Harmful-Resistant Null-Space.
>
> [2] Nullu: Mitigating Object Hallucinations in Large Vision-Language Models via HalluSpace Projection.
>
> [3] Null space-based behavioral control applied to a formation of two quadrotors transporting a cable suspended load.
>
> [4] Outlier Detection through Null Space Analysis of Neural Networks.
>
> - **The difference in using null space between KORE and AlphaEdit:**
>     - **Multimodal null space:** AlphaEdit uses 100,000 text triplets from Wikipedia to extract its null space, only focusing on the purely textual domain. In stark contrast, KORE utilizes a minimal set of **only 256 QA pairs** from the OneVision dataset to extract the **multimodal null space** required to protect prior knowledge and capabilities. KORE achieves excellent knowledge retention performance with a significantly smaller data volume (details in Section 4.1, Table 1). Furthermore, KORE adopts the null-space concept only after **rigorous experimental validation** in the multimodal setting, ensuring it is not merely a blind imitation (details in Section 3.3, Figure 4, and Appendix D).
>     - **Simpler constraints:** AlphaEdit requires null-space constraints during the optimization of every edited knowledge instance **($W + \triangle P$)**, whereas KORE only requires a null-space constraint during the initialization of the LoRA's low-rank matrices $A$ and $B$ **($W_0 + BA$)**. **This merely defines a fine-tuning direction that minimizes interference with prior knowledge throughout the subsequent optimization**. Consequently, KORE's operation is simpler and more convenient.
>     - **Better scalability and flexibility:** AlphaEdit uses text triplet data from Wikipedia, while KORE uses QA pairs from the OneVision dataset, which imposes fewer data format constraints. Furthermore, **KORE can easily implement specific knowledge-oriented constraints based on user needs** (Section 4.2, Page 7-8, Line 368-374+392-402), whether for specific capabilities or factual knowledge, demonstrating extremely strong scalability and flexibility.
>     - **Synergistic optimization objectives:** AlphaEdit primarily focuses on using the null space to constrain the interference with prior knowledge during new knowledge updates, with less attention paid to the new knowledge update performance. However, both knowledge editing and knowledge injection require synergistically optimizing the goals of learning new knowledge (knowledge adaptation) and preserving old knowledge (knowledge retention). **KORE, using the knowledge-oriented approach as its pivot, synergistically optimizes the balance between knowledge adaptation and retention at different stages of knowledge injection**, which better aligns with practical application requirements.
>
> - **Summary of our contribution:** KORE is the first synergistic method of knowledge-oriented augmentations and constraints for injecting new knowledge into large multimodal models while preserving old knowledge. KORE-Augmentation (for accurate adaptation) and KORE-Constraint (for powerful retention) use the knowledge-oriented approach as a pivot to optimize the balance between knowledge adaptation and retention at different stages of knowledge injection. This synergistic design is necessary to address the challenges of poor generalization and catastrophic forgetting (in teaser, Page 1, Figure 1) in multimodal knowledge injection scenarios, which prior continual learning and methods like AlphaEdit cannot solve alone.

---

> ### Author Response · Authors · 2025-11-25
> **Response to reviewer WzoY (2/2)**
>
> > **W2+Q1: Experiments are conducted only on the EVOKE benchmark, while another equally important benchmark in this field, CoIN , is neglected.**
>
> We will clarify from the following aspects:
>
> - As indicated by our teaser and title, our work primarily focuses on factual knowledge injection (e.g., 'Geoffrey Hinton + Nobel Prize + October 2024' and 'Donald Trump + Assassination + July 2024'). This knowledge is unknown to models released in 2023, like LLaVA-v1.5, and we aim to teach it via knowledge injection. **Since EVOKE is the first open-source multimodal knowledge injection benchmark we surveyed**, our experiments are conducted on it to ensure fair and direct comparison with prior continual learning methods. CoIN is a widely used benchmark in the **continual instruction tuning domain**. Since it primarily focuses on continual learning across diverse instruction tasks (e.g., Classification, grounding, VQA, etc.), we do not choose to evaluate on the CoIN benchmark.
> - While we recognize the value of the CoIN benchmark, KORE method requires the deep, structured augmentation of coarse-grained factual text into Multi-rounds of Dialogue Data and Instruction Tasks Data. In contrast, CoIN's data sources (e.g., ImageNet, VQAv2) consist mainly of **simple QA forms that are already fine-grained, making them difficult to adapt to KORE-Augmentation**. Therefore, under the premise of not changing the method's design, we cannot evaluate on the CoIN benchmark, and we hope for your understanding.
>
>
> > **Q2: SEFE consists of two components, ASD and RegLoRA. Since the authors of SEFE did not apply ASD to the EVOKE dataset, did you fully reproduce ASD on the EVOKE benchmark for comparison, or are your SEFE results based solely on its RegLoRA component?**
>
>
> We will clarify from the following aspects:
>
> - **Yes**, since the authors of SEFE did not apply ASD to the EVOKE dataset, the SEFE results in our paper are based solely on its RegLoRA component.
> - Since the RegLoRA-only setting for SEFE introduces unfairness and may mislead readers regarding its performance, **we strictly follow the ASD procedure from the SEFE paper, using GPT-4o on EVOKE to construct EVOKE-ASD** (five versions: X=10%,20%，40%，60%，80%). This ensures fair comparison while adding a new SEFE-compatible (EVOKE-ASD) dataset to the continual learning field. We will subsequently release EVOKE-ASD on an open-source platform.
>
> - The table below presents the performance of SEFE utilizing EVOKE-ASD.
>     - We find that SEFE, when using EVOKE-ASD, shows superior performance in both knowledge adaptation and retention compared to the RegLoRA-only setting.
>     - Furthermore, the overall average score of EVOKE-ASD-enhanced SEFE surpasses EWC, LwF, MoELoRA, O-LoRA, and CIA. This indicates that SEFE is an advanced continuous learning method.
>     - We use the SEFE(ASD-40%) results to update Tables 1 (Page 7) and 2 (Page 8). We hope these efforts meet your approval and prevent readers from misunderstanding SEFE's performance.
>
>
>
>
> > **Table 1: Performance of knowledge adaptation and retention evaluations.**
>
> | Method | K.A↑ | CEM↑ | F1↑ | K.R↑ | COM↑ | OCR↑ | M-DIS↑ | INS↑ | M-IDU↑ | MAT↑ | HAL↑ | Avg↑ |
> | :--- | :---: | :---: | :---: | :---: | :---: | :---: | :---: | :---: | :---: | :---: | :---: | :---: |
> | Replay | 14.67 | 11.36 | 17.98 | 42.69 | 59.72 | 37.98 | 48.64 | 62.33 | 19.31 | 19.17 | 51.67 | 28.68 |
> | EWC | 17.46 | 15.49 | 19.42 | 33.20 | 49.42 | 32.88 | 45.46 | 29.79 | 13.36 | 18.00 | 43.50 | 25.33 |
> | LwF | 17.29 | 14.58 | 19.99 | 33.94 | 53.14 | 28.77 | 43.41 | 36.19 | 13.68 | 18.22 | 44.18 | 25.61 |
> | MoELoRA | 9.33 | 6.45 | 12.20 | 38.50 | 60.79 | 38.79 | 48.27 | 35.03 | 17.85 | 18.79 | 49.99 | 23.91 |
> | O-LoRA | 9.26 | 6.44 | 12.08 | 39.08 | 61.47 | 40.91 | 48.07 | 34.85 | 17.28 | 19.87 | 51.12 | 24.17 |
> | CIA | 17.39 | 14.50 | 20.27 | 33.26 | 52.47 | 33.80 | 45.09 | 34.07 | 10.40 | 12.50 | 44.52 | 25.32 |
> | SEFE(ASD-0%)| **15.13** | 13.38 | 16.88 | **27.30** | 42.06 | 20.43 | 40.17 | 17.73 | 13.25 | 18.20 | 39.30 | **21.22** |
> | SEFE(ASD-10%) | **15.58** | 11.99 | 19.16 | **35.76** | 43.55 | 45.60 | 48.57 | 44.93 | 12.67 | 17.72 | 37.26 | **25.67** |
> | SEFE(ASD-20%) | **17.37** | 13.76 | 20.98 | **34.62** | 40.70 | 44.12 | 48.61 | 45.01 | 12.85 | 18.00 | 33.05 | **25.99** |
> | SEFE(ASD-40%) | **17.98** | 14.12 | 21.84 | **34.38** | 40.03 | 41.28 | 48.88 | 47.16 | 13.48 | 18.18 | 31.67 | **26.18** |
> | SEFE(ASD-60%) | **18.46** | 14.54 | 22.38 | **32.60** | 36.67 | 36.25 | 49.01 | 43.13 | 13.13 | 18.90 | 31.09 | **25.53** |
> | SEFE(ASD-80%) | **20.17** | 16.22 | 24.12 | **32.12** | 36.77 | 35.94 | 48.55 | 42.90 | 10.55 | 18.59 | 31.55 | **26.15** |
> | **KORE** | 35.96 | 30.65 | 41.26 | 38.22 | 52.41 | 40.98 | 48.68 | 38.54 | 16.58 | 18.59 | 51.75 | 37.09 |
>
>
>
> **Hope our clarification could resolve your concern.**

---

> ### Author Response · Authors · 2025-11-28
> **Please tell us if any concern remains**
>
> Dear Reviewer WzoY, ﻿
> ﻿
>
>
> We truly appreciate the effort you have invested in reviewing this paper. We have submitted a **detailed rebuttal** addressing each of your points.
>
>
> ﻿**If you find that any explanation is still insufficient or could benefit from further clarification, please tell us and we would be happy to elaborate.**
> ﻿
>
>
> Thanks for your consideration.
>
>
> ﻿Authors of Paper 374

---

### Official Review · Reviewer_WrUD · 2025-10-31

**Soundness:** 3
**Presentation:** 3
**Contribution:** 2
**Rating:** 6
**Confidence:** 3

**Summary:**

The paper investigates the continual learning in Large Multi-Modal Models. To solve the balance of learning new knowledge and knowledge retention, they propose a KOPE. Extensive experiments demonstrate that KOPE achieves new knowledge injection performance and mitigates catastrophic forgetting.

**Strengths:**

The proposed method sounds reasonable, which makes new knowledge structured, and the covariance matrix keeps previous knowledge.

The experiment in the paper is sufficient, including multiple MLLMs and multiple downstream tasks.

**Weaknesses:**

The survey is insufficient, e.g., related works and baselines have few papers in 2025.

For Figure 5, why is Full-FT lower than KOPE? Theoretically, full fine-tuning is an upper bound for any method.

During the training, how to split the dataset into new knowledge datasets and old knowledge datasets?

**Questions:**

See Weakness.

---

> ### Author Response · Authors · 2025-11-25
> **Response to reviewer WrUD (1/2)**
>
> Dear Reviewer WrUD:
>
> Thank you for your positive feedback and valuable suggestions. We sincerely appreciate the time and effort you have dedicated to reviewing our work. Below, we meticulously provide responses to each of your comments and outline the modifications based on your suggestions. All revisions are highlighted in blue.
>
> ---
> > **W1: The survey is insufficient, e.g., related works and baselines have few papers in 2025.**
>
>
> Thank you for your suggestion. We rewrite the Related Works section to include discussion of relevant 2025 literature and add a comparison with the CIA (ICML'25) method.
> - We revise the `Knowledge Forgetting` section of the related works by discussing 2025 literature relevant to continual learning and knowledge injection[1,2,3,4,5,6,7,8].(Page 3, Line 117-130) This update clarifies their relationship, specifically by asserting that knowledge injection is fundamentally a continual learning problem.
> - We include the **ICML'25 method, CIA**, for comparison with KORE, and the specific results are shown in the table below.
>     - From Table 1, we observe that **KORE performs better than CIA on nearly all evaluations and surpasses it by 11.77 in the Avg score**.
>     - From Table 2, we find that KORE outperforms CIA on all 8 fine-grained knowledge types, indicating that **KORE possesses better knowledge adaptation capability**.
>     - From Table 3, we find that KORE outperforms CIA on all 11 fine-grained knowledge retention metrics, indicating that **KORE possesses better knowledge retention capability**.
>     - **Overall, the addition of the 2025 baselines does not affect the experimental conclusions of the paper**.
>
>
> > **Table 1: Performance of KORE in knowledge adaptation and retention compared with CIA.**
>
> | Method | K.A↑ | CEM↑ | F1↑ | K.R↑ | COM↑ | OCR↑ | M-DIS↑ | INS↑ | M-IDU↑ | MAT↑ | HAL↑ | Avg↑ |
> | :--- | :---: | :---: | :---: | :---: | :---: | :---: | :---: | :---: | :---: | :---: | :---: | :---: |
> | CIA | 17.39 | 14.50 | 20.27 | 34.02 | **52.47** | 33.80 | 45.09 | 34.07 | 10.40 | 17.80 | 44.52 | 25.70 |
> | KORE | **35.96** | **30.65** | **41.26** | **38.22** | 52.41 | **40.98** | **48.68** | **38.54** | **16.58** | **18.59** | **51.75** | **37.09** |
>
> > **Table 2: Performance comparison between KORE and CIA on fine-grained knowledge types.**
>
> | Method |  | News |  |  |  |  |  |  |  | Entity   | | |  |  |  |  |
> |:---|:---:|:---:|:---:|:---:|:---:|:---:|:---:|:---:|:---:|:---:|:---:|:---:|:---:|:---:|:---:|:---:|
> |  |  | Politics |  | Sports |  | Business |  | Health |  | Celebrity |  | Film |  | Album |  | WrittenWork |  |
> |  |  | CEM↑ | F1↑ | CEM↑ | F1↑ | CEM↑ | F1↑ | CEM↑ | F1↑ | CEM↑ | F1↑ | CEM↑ | F1↑ | CEM↑ | F1↑ | CEM↑ | F1↑ |
> | CIA |  | 10.27 | 14.85 | 20.56 | 23.53 | 18.08 | 25.46 | 17.41 | 25.39 | 12.45 | 23.21 | 11.59 | 17.15 | 9.57 | 8.72 | 11.86 | 11.72 |
> | KORE |  | **23.83** | **32.31** | **46.19** | **50.38** | **34.69** | **45.74** | **33.20** | **45.23** | **27.79** | **42.61** | **26.93** | **34.05** | **16.52** | **29.54** | **28.81** | **43.05** |
>
> > **Table 3: Performance comparison between KORE and CIA on fine-grained knowledge retention evaluations.**
>
>
> | Method | MME | MMBench | SEEDBench2Plus | OCRVQA | ScienceQA | MMMU | MIA-Bench | MMDU | MathVista | MathVision | POPE | HallusionBench | Avg |
> | :--- | :---: | :---: | :---: | :---: | :---: | :---: | :---: | :---: | :---: | :---: | :---: | :---: | :---: |
> | CIA | **50.65** | 54.30 | 33.29 | 34.31 | 67.28 | 22.90 | 34.07 | 10.40 | 24.00 | 11.60 | 79.29 | 9.75 | 35.10 |
> | KORE | 49.84 | **54.98** | **37.73** | **44.24** | **68.06** | **29.30** | **38.54** | **16.58** | **25.10** | **12.09** | **80.99** | **22.51** | **40.00** |
>
>
> [1] Continual Learning for VLMs: A Survey and Taxonomy Beyond Forgetting.(2025)
>
> [2] When continue learning meets multimodal large language model: A survey.(2025)
>
> [3] Injecting Domain-Specific Knowledge into Large Language Models:A Comprehensive Survey. (EMNLP2025)
>
> [4] Towards Lifelong Learning of Large Language Models: A Survey.(ACM Computing Surveys2025)
>
> [5] Large Continual Instruction Assistant.(ICML25)
>
> [6] AlphaEdit: Null-Space Constrained Knowledge Editing for Language Models.(ICLR25)
>
> [7] LoRASculpt: Sculpting LoRA for Harmonizing General and Specialized Knowledge in Multimodal Large Language Models.(CVPR25)
>
> [8] LoRI: Reducing Cross-Task Interference in Multi-Task Low-Rank Adaptation.(COLM25)

---

> ### Author Response · Authors · 2025-11-25
> **Response to reviewer WrUD (2/2)**
>
> > **W2: For Figure 5, why is Full-FT lower than KORE? Theoretically, full fine-tuning is an upper bound for any method.**
>
> We will clarify from the following aspects:
>
> - First, we clarify the content of Figure 5: **it compares the CEM and F1-score performance of various methods for knowledge adaptation across 8 fine-grained knowledge types**: Politics, Sports, Business, Health, Celebrity, Film, Album, and Written Work.(The fine-grained type classification references EVOKE[1].)
> - **Full-FT exhibits the flaw of overfitting:**
>     - The paper's teaser (Page 1, Figure 1) illustrates the overfitting flaw of Full-FT: when we expect the concise answer 'Thomas Matthew Crooks', the model instead reproduces knowledge from the training data. This indicates severe overfitting, suggesting the model merely **memorizes data mechanically** without achieving true knowledge internalization. `This phenomenon is also emphasized in the Introduction (Page 2, Line 71-74).`
>     - Appendix F (Page 26-27, Figure 10) provides the training loss curves, showing that **Full-FT's loss decreases to near zero**. Yet, performance comparison in Figure 5 reveals that Full-FT does not successfully inject new knowledge, which is an indication of extremely severe overfitting.
> - **Prior work[1,2,3,4] demonstrates that Full-FT is not the upper bound for performance**, as previous studies show that data augmentation and knowledge augmentation successfully boost knowledge injection capabilities beyond Full-FT.
> - KORE achieves accurate knowledge adaptation via knowledge-oriented augmentation by proposing an automated method that augments knowledge into profound and structured representations. This helps the model achieve high accuracy, which is why Full-FT's performance is weaker than KORE's in Figure 5.
>
> [1] When Large Multimodal Models Confront Evolving Knowledge: Challenges and Pathways.
>
> [2] Physics of language models: Part 3.1, knowledge storage and extraction.
>
> [3] A survey on data synthesis and augmentation for large language models.
>
> [4] New News: System-2 Fine-tuning for Robust Integration of New Knowledge.
>
>
> ---
> > **W3: During the training, how to split the dataset into new knowledge datasets and old knowledge datasets?**
>
> We will clarify from the following aspects:
>
>
> - We first clarify that the **new knowledge dataset is KORE-74K**, derived by applying KORE-Augmentation to EVOKE[1], which serves as the data for knowledge injection. The **old knowledge dataset**, used to protect prior knowledge and abilities, **is obtained by randomly sampling 256 entries from the OneVision[2] dataset**.
> - **New knowledge dataset:** In Section 3.1 (Page 3-4, Line 152-205) and Appendix H.1 (Page 30-31, Line 1568-1634), we detail how the conversion from EVOKE to KORE-74K is achieved. KORE-74K ultimately serves as the new knowledge dataset during the training process.
> - **Old knowledge dataset:** In Section 3.3 (Page 5-6, Line 256-286), we detail that the data used to protect prior knowledge is obtained by randomly sampling 64 entries from each of the four OneVision dimensions: General `VizWiz` , Doc/Chart/Screen `FigureQA`, Math/Reasoning `Geometry3K`, and General OCR `iiit5k`. The final 256 aggregated entries constitute the old knowledge dataset for training. To improve clarity, we add Appendix I and Figure 14 (Page 37) to the manuscript, fully illustrating this construction process.
> - We promise to subsequently release the new knowledge dataset and the old knowledge dataset used in training onto an open-source platform.
>
>
> [1] When Large Multimodal Models Confront Evolving Knowledge: Challenges and Pathways.
> [2] LLaVA-OneVision: Easy Visual Task Transfer.
>
>
> **Hope our clarification could resolve your concern.**

---

> ### Author Response · Authors · 2025-11-28
> **Please tell us if any concern remains**
>
> Dear Reviewer WrUD, ﻿
> ﻿
>
>
> We truly appreciate the effort you have invested in reviewing this paper. We have submitted a **detailed rebuttal** addressing each of your points.
>
>
> ﻿**If you find that any explanation is still insufficient or could benefit from further clarification, please tell us and we would be happy to elaborate.**
> ﻿
>
>
> Thanks for your consideration.
>
>
> ﻿Authors of Paper 374

---

### Author Response · Authors · 2025-11-25
**General Response**

Dear Reviewers,

**We sincerely appreciate your time, efforts, and insightful feedback on our work! We are delighted that all reviewers recognized the motivation, novelty, presentation, and experimental effectiveness of our study.**


We thank reviewers $\text{WrUD(R1)}$, $\text{WzoY(R2)}$, $\text{Uqia(R3)}$ for their insightful feedback. The reviewers praise the proposed method for being reasonable, as it makes new knowledge structured and effectively preserves previous knowledge using the covariance matrix ($\text{R1}$). The KORE-Augmentation is highlighted as a highly innovative, reasonable, and effective data augmentation approach ($\text{R2}$). Reviewers acknowledge that the paper presents strong empirical results, achieving state-of-the-art performance ($\text{R2}$). Furthermore, the experiments are deemed sufficient, encompassing multiple Large Multimodal Models (LMMs) and multiple downstream tasks ($\text{R1}$). Reviewers also commend the paper's overall presentation, noting that it is visually well-presented with clear figures and tables, and the writing is coherent and easy to follow ($\text{R3}$).


Below, we provide point-by-point responses to your comments and outline the revisions made to the manuscript based on your suggestions. All revisions are highlighted in blue. We have summarized our response as follows:


- **Clarification:**
    - Originality and novelty of KORE. ($\text{R2,R3}$)
    - Effectiveness of KORE does not come from the distillation of GPT-4o. ($\text{R3}$)
    - KORE uses knowledge-oriented control as its pivot and is a method capable of synergistically optimizing objectives. ($\text{R3}$)
- **New experiments:**
    - One new baseline for continual learning: CIA(ICML25) ($\text{R1}$)
    - Constructing EVOKE-ASD dataset and updating the performance of SEFE method. ($\text{R2}$)


We warmly encourage you to review the results in the revised manuscript. Hope our response could address your concerns!

Furthermore, please allow us to reiterate the key contribution of our work: **KORE uses knowledge-oriented control as its pivot to synergistically optimize the balance between knowledge adaptation and retention at different stages of knowledge injection, thereby addressing the challenges of poor generalization and catastrophic forgetting.**

- The first synergistic method to address the challenges of poor generalization and catastrophic forgetting in knowledge injection.
- A knowledge-oriented augmentation method that automatically transforms arbitrary knowledge into a profound and structured form.
- A knowledge-oriented constraint method capable of achieving powerful retention performance using only a small amount of data.
- Diverse and complete experimental verification:
    - **Excellent overall performance:** KORE surpasses 9 baselines on overall performance. `Page 5, Table 1`
    - **Potential for model size expansion:** KORE has a greater performance advantage over the optimal baseline on LLaVA-v1.5-13B. `Page 8, Table 4`
    - **Not dependent on the model framework:** Both LLaVA-v1.5 and Qwen2.5-VL demonstrate powerful performance. `Page 8, Table 4`
    - **Extremely scalable:** KORE achieves personalized knowledge retention according to any needs. `Page 8, Table 6, Figure 6`
    - **Superior to general augmentation methods:** KORE Augmentation surpasses four general augmentation methods on knowledge adaptation and retention evaluations. `Page 9, Table 6`
    - **Low rank and low parameters still result in superior performance:** KORE (rank=64, params=108M) surpasses Replay (rank=128, params=304M) on overall performance. `Page 9, Figure 7`


We believe these contributions are crucial for advancing the field of knowledge injection, and we are truly grateful for your recognition of its significance.

Once again, we deeply appreciate the time and expertise you have shared with us. Your encouraging feedback motivates us to continue advancing this work for the broader community, and we are more than happy to add clarifications to address any additional recommendations and reviews from you！

Best regards,

Authors of Paper 374

---

### Author Response · Authors · 2025-11-30
**Summary for AC Consideration**

Dear Area Chair,

We sincerely appreciate your time and effort in handling our submission.

Unfortunately, we receive no reply from any reviewer during the discussion period. However, we believe the **positive scores** provided by **Reviewer WrUD** sufficiently indicate their approval of our work.

Although **Reviewers WzoY and Uqia** give negative scores, **we believe our rebuttal resolves their concerns**. Below, we provide a concise response to the primary worries raised by **Reviewers WzoY and Uqia**:



**(1) Originality and novelty of KORE.** **(Reviewers WzoY and Uqia)**: We prove that KORE's design is innovative and effective from the following perspectives.
- Null space is a widely recognized standard tool.
- Multimodal null space; Better scalability and flexibility; Synergistic optimization objectives.
- KORE is the first synergistic method of knowledge-oriented augmentations and constraints for injecting new knowledge into large multimodal models while preserving old knowledge.




**(3) performance improvement does not come from distilling GPT-4o** **(Reviewer Uqia)**:
- GPT-4o merely acts as the executor of KORE-Augmentation and is not the provider of the knowledge itself.
- Human studies prove that augmented data does not introduce external knowledge.
- General augmentation methods can also achieve performance improvements, but they are far inferior to KORE.




The corresponding clarifications and new experiments are summarized in our **General Response comment below** for your convenience.

Thank you again for your consideration.

Best regards,

Authors of Paper 374

---

### Meta-Review · Area_Chair_3Ra5 · 2026-01-07

**Summary:**

The paper proposes KORE-Augmentation and KORE-Constraint as two components of their approach for knowledge injection in large multimodal models. While the reviewers acknowledge that the framework is reasonable and effective, they raised concerns about the novelty of KORE-Constraint in light of previous work such as AlphaEdit, as well as the lack of experiments on the CoIN dataset. Although the rebuttal addressed some of the other concerns raised by Reviewers WzoY and Uqia, the main issues regarding novelty and the absence of experiments remain insufficiently addressed.

**Reviewer Concerns:**

In my opinion, the concern regarding the similarity between KORE-Constraint (the proposed method) and AlphaEdit is not sufficiently addressed in the author’s response. Both KORE-constraint and AlphaEdit are used for knowledge injection and editing—KORE-constraint is applied to MLMs, while AlphaEdit is applied to LLMs. However, the authors’ explanation does not convincingly articulate the differences between the two methods; in fact, they appear to be very similar, with KORE-constraint introducing only minor changes. Furthermore, the authors did not include AlphaEdit as a baseline for comparison in their experiments, which they should have done if they believe the two methods are substantially different.

Additionally, the concern about the lack of experiments on the CoIN dataset is not adequately addressed. The authors’ response simply acknowledges that their KORE-Augmentation method would not be effective on this dataset, without providing further justification or alternative experiments.

**Reviewer Scores:**

In my opinion, Reviewer WrUD, who initially gave the paper a positive rating of 6, is likely to maintain their positive rating after considering the author’s response.

Reviewer WzoY, who initially gave a negative rating of 4, is also likely to maintain their rating, as I do not think their concerns were sufficiently addressed by the authors—particularly regarding the distinction between KORE-Constraint (the proposed approach) and AlphaEdit, as well as the lack of experiments on the CoIN dataset.

Similarly, Reviewer Uqia, who also gave a negative rating of 2 and shared concerns about the distinction with AlphaEdit, is likely to keep their negative rating.

---

### Decision · Program_Chairs · 2026-01-26

Reject